# WEBCANVAS: BENCHMARKING WEB AGENTS IN ONLINE ENVIRONMENTS

## ABSTRACT

For web agents to be practically useful, they must adapt to the continuously evolving web environment characterized by frequent updates to user interfaces and content. However, most existing benchmarks only capture the *static* aspects of the web. To bridge this gap, we introduce `WebCanvas`, an innovative online evaluation framework for web agents that effectively addresses the dynamic nature of web interactions. `WebCanvas` contains three main components to facilitate realistic assessments: (1) A novel evaluation metric which reliably capture critical intermediate actions or states necessary for task completions while disregarding noise caused by insignificant events or changed web-elements. (2) A benchmark dataset called Mind2Web-Live, a refined version of original Mind2Web static dataset containing 542 tasks with 2439 intermediate evaluation states; (3) Lightweight and generalizable annotation tools and maintenance pipelines that enables the community to collect and maintain the high-quality, up-to-date dataset. Building on `WebCanvas`, we open-source a baseline agent framework with extensible modules for reasoning, providing a foundation for the community to conduct online inference and evaluations. Our best-performing agent achieves a task success rate of 23.1% and a task completion rate of 48.8% on the Mind2Web-Live test set. Additionally, we analyze the performance discrepancies across various websites, domains, and experimental environments. We encourage the community to contribute further insights on online agent evaluation, thereby advancing this field of research.

## 1 INTRODUCTION

The enhanced reasoning capabilities of foundational models (Ouyang et al., 2022; Achiam et al., 2023; Touvron et al., 2023a;b; Liu et al., 2024a; Bai et al., 2023) demonstrate the potential for autonomous agents performing on navigation and information retrieval tasks in real-time within web environment, thereby augmenting the human workforce (Shi et al., 2017; Nakano et al., 2021). However, the journey towards autonomous web agents delivering accurate, robust, fast, and cost-effective outcomes to end-users remains fraught with challenges. These include the inherent scarcity of data, the lack of knowledge and reasoning abilities of high-level actions on certain websites, and the absence of accurate and effective process feedback during execution, among others (Gür et al., 2023; Gur et al., 2024). We posit that a significant barrier to realizing the value of web agents is the lack of an easy-to-use platform for the community to drive effort towards real-time data gathering and web agent online benchmarking. This belief is grounded in following observations.

Digital agents require environmental observations and feedback for context. Thus, dynamic, real-world environments are essential for agent evaluation and data collection. The Internet itself emerges as the most extensive arena for the assessment of agents, offering an unparalleled complexity for environmental interaction (Liu et al., 2018; Zhou et al., 2023). However, the rapid evolution of the web environment introduces significant data distribution shifts over time. Figure 2 summarizes three prevalent patterns of changes in web tasks over time. For example, the Mind2Web dataset (Deng et al., 2024), which archives web-based interactions as static HTML snapshots and was released one year ago, shows that more than half tasks(**50.5%**) changed to some extent in their corresponding live websites one year later. This shift may potentially create discrepancies between the offline and online development and evaluation of real-world web agents. In addition, the accumulated knowledge and training data of static websites leads to the saturation of existing benchmarks, making

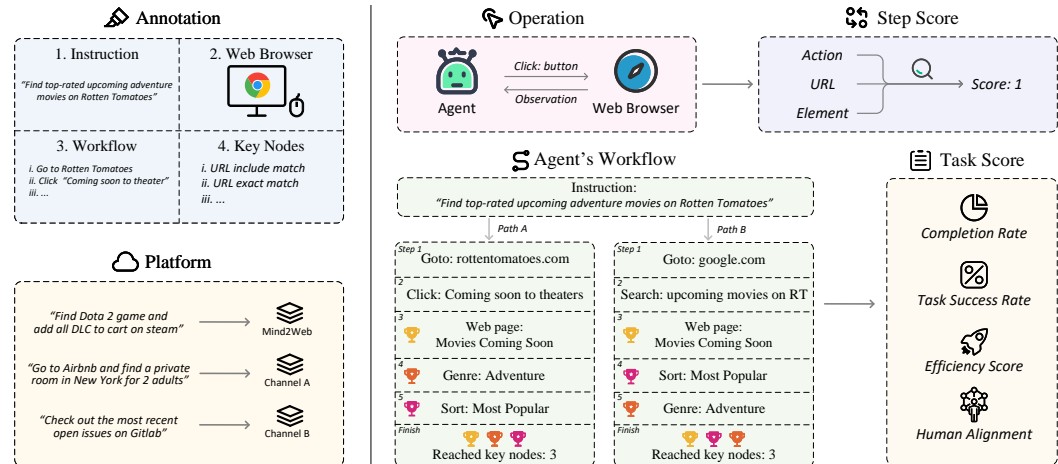

Figure 1: `WebCanvas` framework. The left side depicts the annotation process addressing each task, while the right side demonstrates the evaluation process during inference time, which involves collection of predicted actions, URLs, and elements targeted for interaction in online web environment, allowing for dynamic assessment. The framework accounts for the non-uniqueness of paths in online web interactions, with "Trophies" representing step scores earned upon successfully reaching each key node. Data maintenance pipeline of `WebCanvas` is detailed in §4.2.

it increasingly difficult to compare models and reasoning frameworks fairly and rigorously. We found the MindAct model trained in 2023 outperformed closely-held models like GPT-3.5 (Ouyang et al., 2022) and GPT-4 (Achiam et al., 2023) in Mind2Web static test set, but lagged behind in 2024 online evaluations (§5.1). Although previous works have attempted to evaluate the performance of web agents in online environments through human assessments (Zheng et al., 2024; He et al., 2024), achieving an objective, quantitative, and reproducible evaluation remains challenging.

To bridge this gap, we introduce `WebCanvas`, a dynamic and real-time framework designed for online evaluation of web agents with three key features. (1) **Progress-aware evaluation with key node annotation.** Existing evaluation metrics that focuses on action prediction accuracy (Deng et al., 2024; Zheng et al., 2024) can falsely penalize valid alternative solutions while outcome-based evaluation (Zhou et al., 2023; Koh et al., 2024; Mialon et al., 2023) requires fully reproducible standalone web environments. To address this gap, we introduce a novel concept termed "key nodes" – essential milestones that any task process must traverse, irrespective of the path taken. A comparative illustration of key nodes with these existing methods is provided in Figure 3. Key node annotation allows for a detailed, continuous analysis of agent behaviors, thereby enhancing insight into their decision-making strengths and weaknesses. (2) **Collaborative platform for community-driven annotations.** `WebCanvas` supports recording and annotation of web-based tasks and their corresponding key node evaluation through an advanced recording browser plugin with transparent data access. Furthermore, we have open-sourced an agent reasoning framework that enhances the integration and customization of various agent modules for online web tasks. This initiative provided guidelines and toolkits for the community to effectively scale data for online evaluation within real-world settings in their own scenario. (3) **Cost-effective maintenance to sustain evaluation validity.** Online environment is continuously evolving, making maintaining data validity a challenge. To address this, `WebCanvas` employs an efficient maintenance strategy with scheduled monitoring and automated alerts that quickly identify action sequences and key nodes validity. When data shifts occur, our test report with error messages guide data owner through quick and effective data corrections. This approach allows us to dynamically adjust our evaluation sets in response to real-time changes in web content with acceptable cost.

Based on `WebCanvas` framework, we create **Mind2Web-Live** dataset for the community. This dataset contains 542 tasks sampled from Mind2Web (Deng et al., 2024), and we annotate each task with key node verification. Extensive comparisons show that GPT-4 with memory and ReAct (Yao et al., 2023) reasoning achieved the best task success rate of 23.1%. In addition, our online evaluation reveals discrepancies with offline settings, demonstrating that models which perform competitively

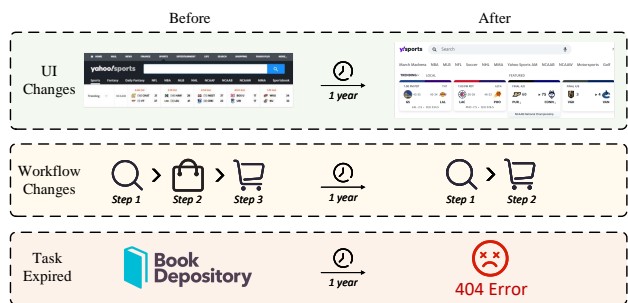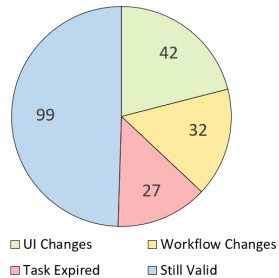

Figure 2: The left side illustrates three typical types of web task evolution over time. We further annotate and classify the status of 200 sampled tasks from Mind2Web training set between May 2023 and May 2024, showing more than half underwent changes to some extent within just one year: 21% experienced UI redesigns, 16% had workflow modifications, and 13.5% became completely expired.

in static offline evaluations do not necessarily maintain their competency in dynamic online environments. We further analyze the impact of various factors specific to online evaluation, such as IP location variability, and suggest maintaining a consistent setup within our framework to ensure reliable results. Finally, we investigate the use of key node annotations as a form of intermediate reward for in-context reasoning. Our findings suggest that web agents can benefit from human-provided reward signal, whereas even advanced models exhibit inaccuracies when generating such intermediate progress indicators without any reference. These inaccuracies subsequently impair execution performance.

## 2 PROBLEM FORMULATION OF INTERACTIVE WEB-BASED TASK

The real-world web environment can be formulated as: $(\mathcal{S}, \mathcal{A}, \mathcal{T}, \mathcal{O})$ with state space $\mathcal{S}$, action space $\mathcal{A}$(Table 10), deterministic transition function $\mathcal{T} : \mathcal{S} \times \mathcal{A} \longrightarrow \mathcal{S}$ and a state observation space $\mathcal{O}$(§5). Given a task instruction $i$, current observation $o_t \in \mathcal{O}$ and the action history $a_{1:t-1}$, an agent issues an action $a_t \in \mathcal{A}$. Consequently, after the execution of the action, the environment transitions to a new state $s_{t+1} \in \mathcal{S}$, and the corresponding observation updates to $o_{t+1} \in \mathcal{O}$. To measure the completion of tasks, we have defined key nodes and evaluation metrics, which are elaborated in §3.1 and §3.2.

## 3 WEBCANVAS: AN ONLINE EVALUATION FRAMEWORK FOR WEB AGENTS

### 3.1 DEFINITION OF KEY NODES

The concept of "key nodes" is one of the pivotal ideas in our work. Key nodes refer to indispensable steps in the process of completing specific web tasks, meaning that regardless of the path taken to accomplish a task, these steps are required. These may involve navigation to certain webpages or the performance of specific actions on web pages, such as filling out forms or clicking buttons. This design philosophy not only reflects the dynamic nature of the web environment but also captures the diversity of paths present in real-world web pages.

As illustrated in Figure 1, consider the task of "Find top-rated upcoming adventure movies on Rotten Tomatoes" as an example. Users might start directly at the Rotten Tomatoes homepage or use a search engine to navigate straight to the "New Movies Coming Soon" page of the Rotten Tomatoes. Moreover, when filtering the movies, users might choose to first apply a filter for the "adventure" genre and then sort by popularity, or alternatively, sort by popularity before applying the genre filter. Despite the availability of different paths to achieve the goal, entering the specific page and performing the genre and popularity sorting are essential steps in accomplishing the task. Therefore, these three steps are identified as "key nodes".

In the dynamic and noisy real-world web environment, identifying these key nodes is challenging due to the potential changes in page content and UI updates, which could render element selector paths obsolete. Therefore, we preferred to use URL state as identifiers for key nodes rather than element interaction, which enhanced the Benchmark's robustness against layout changes. Only element class

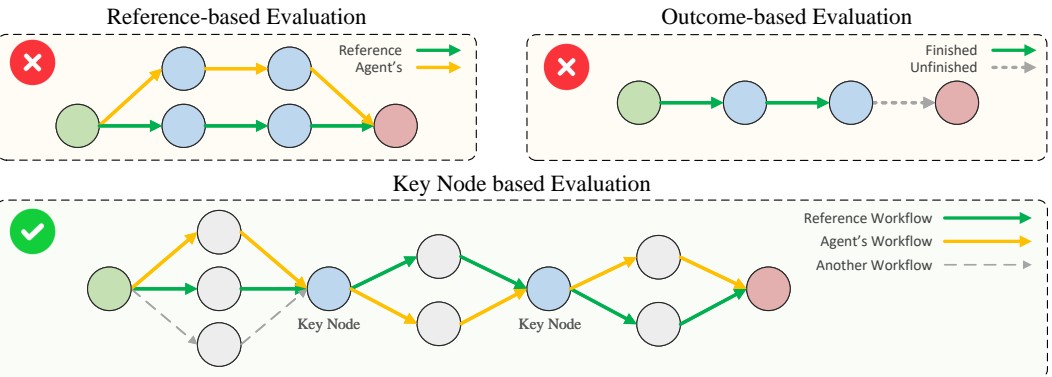

Figure 3: Comparison of different evaluation methods.

methods are considered for key nodes that cannot be represented by URLs. The detailed judgment method is described in Appendix C. By defining key nodes, `WebCanvas` is able to dynamically assess the execution capabilities of web agents in real-world web environments, offering a practical and flexible evaluation method for the development of web agents.

## 3.2 EVALUATION METRICS

The evaluation metrics of `WebCanvas` comprised of two main components: step score and task score. The step score evaluates the agent's performance with regard to each key node, defining three types of evaluation targets along with three evaluation functions at each step. The task score includes two functions to assess the task's completeness and overall execution efficiency.

**Step Score**   Inspired by previous works (Zhou et al., 2023; Koh et al., 2024), we introduced three evaluation targets in calculating step score, allowing us to examine from different aspects: **URL**, **Element Path**, and **Element Value**. We implemented three match functions for these targets: **Exact Match**, **Include Match**, and **Semantic Match**. Each key node is associated with an evaluation function, which comprises an evaluation target and a match function. One step score is awarded when the agent successfully reaches a key node and passes the associated evaluation function verification. Table 1 shows a list of applicable evaluation functions and their introductions for reference. Some examples are shown in Table 7. We use **Completion Rate** to represent the MICRO average of Step Scores.

Table 1: Overview of evaluation functions. "E" is short for Web Element.

| Match Function | Description | URL | E. Path | E. Value |
|---|---|:---:|:---:|:---:|
| Exact Match | Precise matching, such as URL parameters or form fields. | ✓ | ✓ | ✓ |
| Include Match | Evaluates if output includes the reference, ideal for keyword detection. | ✓ | ✗ | ✓ |
| Semantic Match | Uses LLM for complex content reasoning tasks, like product identification. | ✓ | ✗ | ✓ |

**Task Score**   Task performance is evaluated using two main metrics: the **Task Success Rate** and the **Efficiency Score**. The **Task Success Rate** is determined by the proportion of tasks in the test set that are completed successfully, where a task is considered complete if all designated key nodes are achieved. The **Efficiency Score** is calculated using the formula $ES = \frac{L}{P}$. $L$ represents the total number of steps the agent took to complete the task, and $P$ is the cumulative step score obtained by the agent upon completing the task.

## 4 MIND2WEB-LIVE: A REAL-TIME ONLINE BENCHMARK FOR WEB AGENTS

### 4.1 DATASET CONSTRUCTION

To develop a real-world online benchmark for web agents, we introduce Mind2Web-Live, which is derived from tasks present in the Mind2Web (Deng et al., 2024) dataset. We employed `WebCanvas` framework as a guidance for the sampling and re-annotation of these tasks. Consequently, we selectively excluded all tasks that contained time-sensitive descriptions, such as those involving specific dates or times. We randomly sampled 601 tasks from the training set and included all 179 tasks from the cross-task test set, which were then re-annotated in a real-world online environment.

The annotation process presented multiple challenges. Notably, due to updates in website content and operational changes, we discovered 96 tasks that were no longer applicable and subsequently removed them from the dataset. Additionally, 142 tasks were discarded due to ambiguous task definitions, log-in requirements or the difficulty in clearly defining key nodes. To enhance the clarity and reliability of task execution, we revised the instructions for 51 tasks.

For human trajectory and key node annotation, we developed and made public a browser plugin and an annotation platform. After a rigorous annotation and review process, described in Appendix A, 542 high-quality tasks were established for the Mind2Web-Live dataset, including 438 of the training set and 104 of the test set. As shown in Table 2, Mind2Web-Live encompasses 2439 key nodes and 4550 detailed annotation steps. The tasks in the dataset cover a wide range of webpage types and operations, designed to comprehensively evaluate the performance of web agents in a dynamic and variable online environment. The distribution of the evaluation function is illustrated in Figure 5.

### 4.2 DATASET MAINTENANCE

We pay special attention to the dynamic nature of the benchmark to adapt to the constantly changing web environment. We recognize that updates and changes to website content, such as UI updates, database changes, or website close-down, are inevitable as time progresses. Such changes may lead to the obsolescence of previously defined tasks or key nodes.

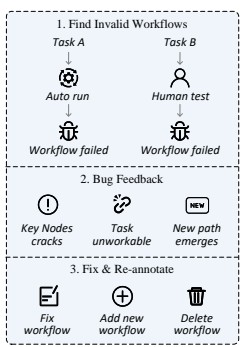

We are thus committed to process a regular data maintenance schedule every two months, as shown in Figure 4. We first developed a community-driven platform where dataset users can visualize details of each task and report any issues through a bug-reporting feature. In addition to community supervision, we leveraged the data stored during the annotation stage to ensure a stable playback of these recorded human trajectories, with any invalidities in the workflows or key nodes being promptly reported. Appendix H provides an example of a report. Suspicious tasks are re-annotated during our reviews to ensure that each task accurately reflects the current web environment.

Figure 4: System of maintenance

Over the past two months, we reviewed 104 tasks in the Mind2Web-Live test set. During this review, we found that 5 tasks had underwent key nodes degradation. The four authors took responsibility for the maintenance work, with each spending less than an hour per maintenance cycle, making this an acceptable cost. For invalid workflows, we updated both the trajectories and the key nodes, a process that usually takes about 5 minutes due to our previous annotation experience. For invalid key nodes, we only needed to update the key node functions with around 2 minutes.

## 5 EXPERIMENT

Inspired by previous work (Yao et al., 2023; Zhou et al., 2023; Zheng et al., 2024), we built a universal baseline agent framework that consists four key modules: Planning, Observation, Memory and Reward. This framework is engineered to be plug and play, operating in real-world online web environments, serving as a foundation for the community to benchmark with rather than introducing new innovations. Detailed implementation is provided in the Appendix E.

Table 2: Data distribution

| Statistic | Number |
|---|---|
| Total selected tasks | 780 |
| - Expired Tasks | 96 |
| - Unable to annotate | 142 |
| - **Mind2Web-Live** | **542** |
|   - training set | 438 |
|   - test set | 104 |
| Annotate steps | 4550 |
| Avg. steps | 8.39 / task |
| Eval functions | 2439 |
| Avg. Eval functions | 4.5 / task |

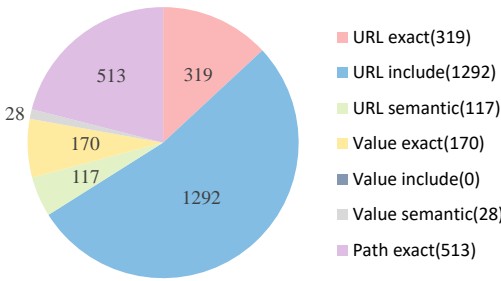

Figure 5: Evaluation Function distribution

Table 3: Comparison of web agent performance in online and offline evaluations. We randomly sampled 40 instances from the Mind2Web-Live test set. These were then tested in both online and offline settings. GPT-3.5 denotes gpt-3.5-turbo-0125, and GPT-4 denotes gpt-4-0125-preview. 'Task SR(0)' and 'Task SR(1)' denote the Task Success Rates with zero tolerance and tolerance for error at one step (or key node), respectively.

| Model | Offline | | | Online | | |
|---|---|---|---|---|---|---|
| | Step SR(%) | Task SR(0)(%) | Task SR(1)(%) | Completion Rate(%) | Task SR(0)(%) | Task SR(1)(%) |
| MindAct | 44.3 | 10.0 | 25.0 | 25.5 | 7.50 | 12.5 |
| GPT-3.5 | 15.5 | 2.50 | 7.50 | 35.4 | 10.0 | 17.5 |
| GPT-4 | 28.4 | 5.00 | 22.5 | 41.1 | 10.0 | 25.0 |

## 5.1 DISCREPANCY BETWEEN OFFLINE EVALUATION AND ONLINE GENERALIZATION

The settings of evaluation on offline datasets that reflect real-world intents, such as Mind2Web (Deng et al., 2024), are inherently different from `WebCanvas` framework. Nevertheless, we managed to study the qualitative discrepancy between offline evaluation and online generalization. During online inference, we attempted to reproduce the original prompt of the MindAct model, which was trained and evaluated on the offline dataset, as proposed in the Mind2Web paper. It is important to note that the evaluation metrics used in offline evaluation differ from those proposed in our online evaluation framework. The Step Success Rate in offline testing assesses the accuracy of single-step action prediction, and for the entire task dimension, a positive reward is given only when all single-step actions are correctly predicted, which is not the case in online evaluation, as we evaluate the intermediate state, not the referenced action.

## 5.2 MAIN RESULT

In our experiments, we observed significant discrepancies between offline and online testing. The results, as detailed in Table 3, show that the model trained on the Mind2Web training set struggles to generalize to the online environment a year later. The comparative performance of MindAct-Large (Deng et al., 2024), GPT-3.5, and GPT-4 in the online environment was opposite to that in offline testing. We further analyzed why such differences occur. Specifically, through human annotation of 100 evaluation steps of GPT-4, we noted that approximately 30% of its actions were reasonable, indicating that these models are possibly capable of generating valid paths under the prompt, albeit penalized by the offline reference-based evaluation method. We also discovered that the MindAct model, when reasoning in an online environment, frequently encountered difficulties in recovering from erroneous states. When entering web pages less relevant to the task goal, the MindAct model had a high probability of outputting null actions, causing the task to terminate.

Furthermore, comparative performance of different models in Table 4 indicates that GPT-4 outperforms other models in both effectiveness and efficiency in web agent tasks within a live environment, with Qwen being the best-performing open-source model. However, there remains considerable room for future enhancements across all models. These results underscore the need for models that can better generalize to dynamic, real-world web environments.

Table 4: Performance of different models without the reward module on the Mind2Web-Live test set, sorted by Completion Rate from highest to lowest. Qualitative analysis of agent performance in online environment are illustrated in Appendix G.

| Model | Completion Rate (%) | Task SR (%) | Efficiency Score |
|---|---|---|---|
| GPT-4-0125-preview | **48.8** | **23.1** | **2.47** |
| Claude-3-Sonnet-20240620 | 47.9 | 22.1 | 2.92 |
| GPT-4o-2024-05-13 | 47.6 | 22.1 | 2.88 |
| Gemini-1.5-Pro | 44.6 | 22.1 | 4.48 |
| GPT-4-turbo-2024-04-09 | 44.3 | 21.2 | 2.78 |
| Claude-3-Sonnet-20240229 | 43.9 | 20.2 | 3.34 |
| Qwen1.5-110B-Chat | 43.9 | 20.2 | 4.02 |
| GPT-4o-mini-2024-07-18 | 42.9 | 21.2 | 2.97 |
| DeepSeek-V2 | 41.2 | 18.3 | 4.44 |
| Qwen2-72B-Instruct | 40.9 | 15.4 | 4.60 |
| Claude-3-Opus-20240229 | 40.3 | 14.4 | 3.52 |
| GPT-3.5-turbo-0125 | 40.2 | 16.3 | 3.03 |
| Mixtral-8x22B | 37.2 | 17.3 | 4.80 |
| Qwen1.5-72B-Chat | 35.6 | 15.4 | 4.29 |
| Gemini-Pro | 35.3 | 13.5 | 4.69 |
| Claude-3-Haiku-20240307 | 33.4 | 16.3 | 4.27 |
| Qwen1.5-7B-Chat | 24.5 | 10.6 | 8.34 |

## 6 ANALYSIS

### 6.1 FACTORS INFLUENCING AGENT PERFORMANCE

In this section, we delve into the factors influencing agent performance across a range of web tasks. Through a series of experiments, we assessed the impact of task complexity, website dynamics, task domain, key node distribution in the dataset, and the experimental setup—including system specifications, browser engine, and IP location.

Our findings reveal that increased task complexity directly correlates with diminished agent performance. The domain of the task also significantly affects performance, with agents handling entertainment-related tasks more adeptly than those involving shopping or travel. This variation suggests LLMs' capacity of semantic understanding and reasoning differs across domains and websites. Moreover, the experimental environment plays a crucial role in agent performance. We recommend experimenting on a Windows platform using Chrome or Firefox browser engines, preferably on servers located in the United States. Statistics and experiment results are detailed in Appendix F.2.

### 6.2 NECESSITY OF KEY NODE EVALUATION IN LIVE ENVIRONMENTS

Previous agent evaluation methods primarily focus on two aspects: reference-based evaluation (Deng et al., 2024; Zheng et al., 2024) and outcome-based evaluation (Zhou et al., 2023; Koh et al., 2024; Mialon et al., 2023). However, these methods falter when applied to the unpredictable nature of live web tasks. To address the inherent variability in task completion paths within an online evaluation framework, we employed Sankey diagrams to visualize the trajectories of our web agent and human demonstrations on tasks where our agent successfully navigated all designated key nodes in Figure 10 within §F.2.

We further annotate Mind2Web-Live test set to identify whether the final key node is a sufficient condition for task completion. It turns out only 46 out of 104 tasks met this criterion. This finding starkly illustrates that solely evaluating the final state or outcome is inadequate for web environments that are not fully reproducible. As shown in Figure 3, key node based evaluation enhances explainability of agent performance, prevents illegal shortcuts taken and facilitates the modeling of structured in-progress rewards, valuable for both in-context reasoning experiments and future reinforcement learning training.

Table 5: Performance of different models with reward module, based on a random sample of 130 tasks from the Mind2Web-Live dataset. "(+)" indicates the inclusion of a reward module with human-labeled reward. Bold numbers represent the best values across different planning models. Model notation follows Table 3, except for gpt-4-vision-preview(GPT-4V). Human Alignment score represents agents' alignment with human decision on task completion, while the larger indicates better alignment, detailed in Appendix D.

| Planning Model | Reward Model | Completion Rate (%) | Task Success Rate (%) | Efficiency Score | Human Alignment |
|---|---|---|---|---|---|
| GPT-3.5 | / | 34.6 | 13.8 | 5.25 | / |
| GPT-4 | / | 46.9 | **16.9** | 3.77 | / |
| GPT-4 | GPT-3.5 | 43.5 | 16.2 | 3.24 | 0.445 |
| GPT-4 | GPT-4 | 42.1 | 13.8 | 3.07 | 0.430 |
| GPT-3.5 | GPT-4 | 36.6 | 10.8 | 3.73 | 0.385 |
| GPT-4 | GPT-4V | 42.4 | 8.5 | 3.42 | 0.419 |
| GPT-3.5 | GPT-4(+) | **43.6** | **13.8** | **3.28** | **0.452** |
| GPT-4 | GPT-4(+) | **52.3** | 12.3 | 3.27 | **0.506** |
| GPT-4 | GPT-4V(+) | 51.3 | 12.3 | **2.71** | 0.502 |

## 6.3 PLANNING WITH HUMAN-LABELED REWARD

Reward modeling for agent tasks is crucial in both in-context reasoning and reinforcement learning (Shinn et al., 2024; Bai et al., 2024). Previous research has proven that LLMs can generate high quality reward signal to enhance reasoning performance across various agent tasks (Shinn et al., 2024; Pan et al., 2024). However, recent research adopting an un-tuned foundation model for self-reward prediction shows that their effectiveness is not consistent in specific domains (Olausson et al., 2023; Shinn et al., 2024). Our preliminary experiments indicate that agent performance do not benefit from a self-reward module in the online web environment. This is attributed to several factors, such as overconfidence in task completion assessments and the long-term impact of poor-quality reward signals accumulated in agent memory. Thus it raises a natural question - *Does the quality of the reward signal hinder the self-reward module's effectiveness in online web environments?* In our study, we introduced a reward module with human-labeled reward. The experimental results on Mind2Web-Live, which confirm our hypothesis, are detailed in Table 5.

From the original data, we extracted post-action URLs, action types, CSS selector paths, and key nodes functions as metadata for our golden reference synthesis. We then employed a carefully designed prompt available in Appendix K, using GPT-4 to generate a structured linguistic guidance for task progress estimation for each task. This guidance includes the overall goal of the task and task completion criteria, specifically highlighting all key nodes for the task to be considered fully completed. We then integrate the content of the current task's golden reference with the original design of history and current observation for reward reasoning. From comprehensive experiments, we find that the integration of a reward module does not enhance agent performance and may even lead to a decline in Task Success Rate and Task Completion Rate. This finding aligns with findings in (Shinn et al., 2024) about the effect of self-reflection modules in web agent tasks. However, we find the Completion Rate improves in both GPT-3.5 and GPT-4 experiments with the integration of a reward module with human-labeled reward, despite the reward module triggering premature stops. These findings point out the importance of better reward modeling in web agent reasoning.

## 7 RELATED WORKS

**Agent Benchmarks** Early researches (Shi et al., 2017) (Liu et al., 2018) provided relatively simple simulations and assessment methods for web navigation tasks. However, with the rise of Large Language Models, these methods have become inadequate for assessing agents' capability. Recent studies have chosen to construct realistic simulated environments (Yao et al., 2022; Zhou et al., 2023; Koh et al., 2024; Drouin et al., 2024), use offline saved datasets (Deng et al., 2024; Lu et al.,

Table 6: Case study of previous benchmarks

| Benchmark | Real-world Intents | Dynamic Environment | Keep Updated | Intermediate Env. State | Easy to Scale | Disk Usage |
|---|---|---|---|---|---|---|
| MiniWoB++ (Liu et al., 2018) | ✗ | ✓ | ✓ | ✗ | ✗ | < 1GB |
| WebShop (Yao et al., 2022) | ✗ | ✓ | ✗ | ✗ | ✗ | ∼ 10GB |
| Mind2Web (Deng et al., 2024) | ✓ | ✗ | ✗ | ✗ | ✗ | ∼ 10GB |
| WebArena (Zhou et al., 2023) | ✓ | ✓ | ✗ | ✗ | ✓ | > 100GB |
| VWebArena (Koh et al., 2024) | ✓ | ✓ | ✗ | ✗ | ✓ | > 100GB |
| GAIA (Mialon et al., 2023) | ✓ | ✓ | ✗ | ✗ | ✓ | < 1GB |
| WEBLINX (Lu et al., 2024) | ✓ | ✗ | ✗ | ✓ | ✗ | < 1GB |
| OmniACT (Kapoor et al., 2024) | ✓ | ✗ | ✗ | ✓ | ✗ | < 1GB |
| WebCanvas | ✓ | ✓ | ✓ | ✓ | ✓ | < 1GB |

2024), or select relatively stable answers to assess the capabilities of web agents (Mialon et al., 2023). In terms of dynamic evaluation methods, many studies (Kiela et al., 2021; Ma et al., 2021; Jain et al., 2024) have proposed their own solutions. Moreover, beyond network platforms, several initiatives have also been undertaken on other platforms such as Android mobile devices, operating systems, and databases (Rawles et al., 2024; Liu et al., 2024b; Xie et al., 2024). We perform a more comprehensive case study on previous web agent benchmarks in Table 6, WebCanvas aims to more comprehensively test agents' capability in the real world through key nodes and corresponding evaluation functions.

**Agent Frameworks** In the area of reasoning frameworks, several studies have achieved notable success in logical reasoning challenges (Wei et al., 2022; Yao et al., 2024; 2023; Shinn et al., 2024; Sumers et al., 2024). Regarding web agent reasoning frameworks, many researches has been conducted to enhance the capabilities of web agents (Nakano et al., 2021; Gur et al., 2024; Gür et al., 2023; Kim et al., 2024; Lo et al., 2023; Lai et al., 2024). Some studies have introduced multi-modal modules that integrate visual and semantic information, thereby enhancing the capabilities of agents on web platforms (Zheng et al., 2024; Furuta et al., 2024; He et al., 2024).

## 8 DISCUSSION & LIMITATIONS

Developing a suitable evaluation framework is a fundamental component in the advancement of autonomous web agents. This research addresses the challenge of live evaluation in a real-world web environment. Among these are the need to define key nodes in a completely open environment, unify the inference processes across different digital autonomous agents, and reduce the maintenance costs associated with real-time data and evaluation functions. Through our efforts, we have made significant strides toward establishing a robust and accurate online evaluation system for web agents.

However, the transition to live, dynamic evaluations in unpredictable online environments introduces new complexities not present in controlled, offline settings. The unsolved challenges we encountered in online evaluation of web agents include network instability, dynamic and complex task pathways, and the limitations of static evaluation functions. These challenges highlight the necessity for ongoing research and community efforts to refine and enhance evaluation frameworks for autonomous web agents in complex, real-world environments. For more details, please refer to Appendix J.

## 9 CONCLUSION

In this work, we have pioneered the development of framework for online evaluation of web agents, and investigated the challenges associated with online evaluation and the difficulties faced by current web agent reasoning frameworks in online inference. Simultaneously, we have constructed toolkits and a community-driven platform that empowers web agent researchers and developers to build datasets and evaluate their web agent frameworks and models in an online environment while collecting feedback on dataset design, data annotation quality, and data validity throughout the process. We strongly encourage further work on online datasets, web agents, and evaluation function designs. By fostering a collaborative and iterative value to dataset creation and evaluation, we eagerly anticipate the continued growth of advancement of autonomous intelligence.

## ETHICS STATEMENT

In this work, all annotators for the Mind2Web-Live dataset were fully informed about the data annotation tasks and compensated for their participation. We take full care of privacy and data security, ensuring that no sensitive personal information was collected, and all tasks can be completed without login.

## REPRODUCIBILITY STATEMENT

We have taken significant measures to ensure the reproducibility of our results. The code used for the experiments, as well as the Mind2Web-Live dataset, are available in the supplementary materials. Additionally, comprehensive guidelines for setting up the environment and the parameters required to run the experiments are included.

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

# A  DATA COLLECTION DETAILS

## A.1  RECORDING PROCESS

In the construction of Mind2Web-Live, the quality and reliability of the data are paramount. To this end, we have employed an efficient tool(Figure 6) for recording browser operations. The tool precisely captures browser interaction from the users, covering a wide range of activities such as clicks and input actions. The recorded details include the type of operation, execution parameters, target element's selector path, element content, and its coordinates on the webpage. Moreover, the tool accompany each step with a webpage screenshot, not only facilitating process replication but also providing a visual reference for workflow validation and review(Figure 18, 19). This approach enables us to comprehensively record all the steps required to complete specific tasks, forming the foundation of Mind2Web-Live. Upon completion of the data recording, we meticulously annotated the key nodes of each process along with their corresponding Evaluation Functions.

## A.2  ANNOTATION PROCESS

In our study, the annotation process plays a pivotal role in ensuring data quality and task validity. To ensure the accuracy and consistency of data annotations, we assembled an annotation team comprised of several authors of this paper and five senior undergraduate students majoring in Computer Science. Not only do the members of the annotation team possess a solid background in Computer Science, but they also received specialized training to ensure consistency in their understanding and identification abilities in annotating key nodes. Prior to beginning the formal annotation process, all annotators were rigorously trained over a period of two weeks, which included trial annotations that were subsequently not included in the final dataset.

During the annotation phase, we employed a comprehensive reward mechanism. Each annotator was compensated based on the number of tasks they completed, with additional bonuses awarded for high-quality annotations to encourage precise and consistent results. This combined reward system not only bolstered work enthusiasm but also enhanced the overall quality of the annotation work, laying a solid foundation for the construction of an efficient web agent benchmark.

To guarantee the quality of annotations, we instituted a variety of strategies. Each task was annotated independently by one annotator, followed by individual reviews by two other members to verify the accuracy of the key nodes. Throughout the annotation process, we regularly organized discussion sessions for the annotation team to share their experiences and challenges encountered, thereby improving the overall efficiency and quality of the team's annotations.

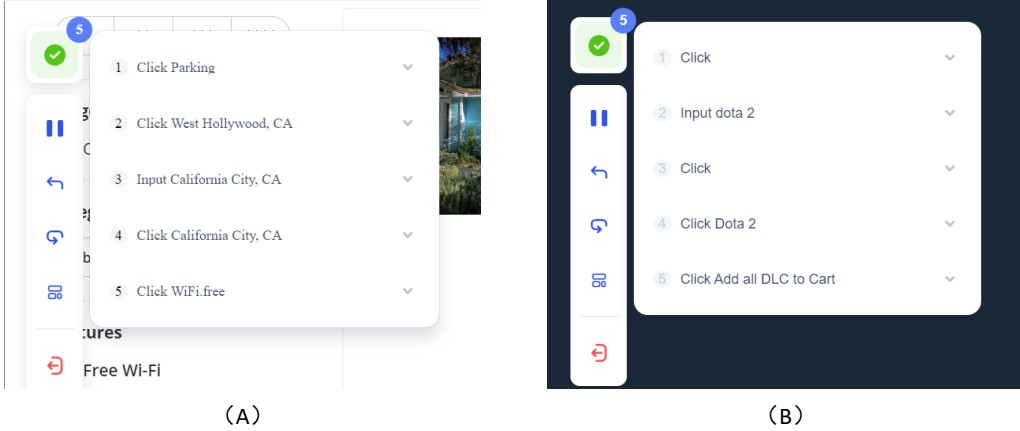

Figure 6: An illustration of the Annotate Tool being used to annotate two distinct tasks: (A) "Find parking in California city for Limos which also offers free wi-fi in yelp", and (B) "Find Dota 2 game and add all DLC to cart in steam".

Table 7: Example annotations of the Evaluation Functions

| State | Title | Annotation Details |
|---|---|---|
|  | Locate a large store in Washington that has kids' and maternity products in uniqlo | Evaluation Function: `Element value semantic match`

Instructions: `Decide Whether is searching for Washington D.C.` |
|  | Find parking in California city for Limos which also offers free wi-fi in yelp | Evaluation Function: `URL include match`

Param: `attrs`
Value: `WiFi.free` |
|  | Find Dota 2 game and add all DLC to cart in steam | Evaluation Function: `Element path exact match`

Selector: `//*[@id="dlc_purchase_action"]/div[2]/a/span` |

## A.3 TASK DISTRIBUTION AND DOMAIN COVERAGE

See Table 8.

Table 8: Task Distribution and Domain Coverage

| Domain | Subdomain | Mind2Web-Live Test | Mind2Web-Live Train |
|---|---|---|---|
| Entertainment | Sports | 9 | 32 |
| | Event | 5 | 20 |
| | Game | 3 | 24 |
| | Movie | 9 | 30 |
| | Music | 5 | 18 |
| | General | 3 | 28 |
| Shopping | Auto | 7 | 33 |
| | Department | 6 | 8 |
| | Digital | 6 | 15 |
| | Fashion | 3 | 15 |
| | Speciality | 13 | 44 |
| Travel | General | 0 | 11 |
| | Airlines | 5 | 18 |
| | Car rental | 1 | 11 |
| | Ground | 9 | 28 |
| | Hotel | 3 | 12 |
| | Restaurant | 6 | 31 |
| | Other | 11 | 60 |
| **Total** | | **104** | **438** |

## B    COMPARISON OF THE MIND2WEB-LIVE AND MIND2WEB DATASETS

Table 9: Comparison of the Mind2Web-Live and Mind2Web Datasets. "Ele." indicates "Element", "Op." indicates "Option" and "SR" indicates "success rate".

| Attributes | Mind2Web-Live | Mind2Web |
|---|---|---|
| Dataset Size | 542 | 2350 |
| Evaluation Environment | Real-world Online | Offline |
| Evaluation State | Key Nodes | Each Step |
| Target Element | Element, URL | Element, Option |
| Evaluation Metrics | Step Score & Task Score | Step(Ele., Op.) SR & Task SR |
| Avg. Steps | 8.39 / task | 7.3 / task |

## C    HOW TO DEFINE EVALUATION FUNCTIONS

**For input operations on the page**    First, determine whether it is a necessary condition for task completion. If it is a necessary condition, then judge whether the execution result can be reflected by the change of the URL. If so, simply take the state after execution as the key node and select the evaluation function as URL exactly/included/semantic match.

If it cannot be reflected by changes in the URL, it needs to be defined as a key node based on click or input operations. Select element path exactly match or element value exactly/included/semantic match for input operations (to determine whether the content of the input element matches).

**For click operations on the page**    Firstly, determine whether it is a necessary condition for completing the task. If it is a necessary condition, then judge whether the execution result can be reflected by the change of the URL. If so, simply take the state after execution as the key node and select the match rule as URL exactly/included/semantic match.

If it cannot be reflected by the change of URL, each click operation should be defined as a key node, and the match can be selected as element element path exactly match or element value match.

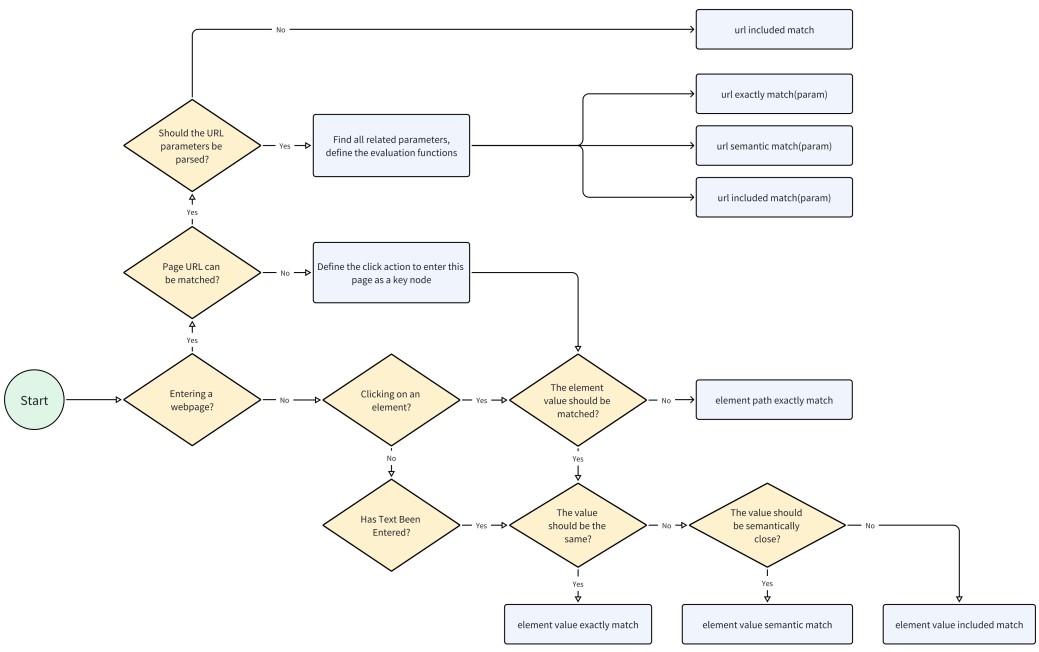

Figure 7: Guidance on how to define an evaluation function for a key node.

## D    ADDITIONAL EVALUATION METRICS

**Human Alignment Score**    The Human Alignment Score(HAS) assesses how well an agent's workflow aligns with human behavior. It's crucial for agents not just to be efficient, but to operate in ways that resemble human actions. The evaluation of this aspect is conducted by contrasting the agent's task completion signal with the ground truth annotations provided by humans, to gauge the level of consistency. An agent that accurately issues a completion signal upon task completion is deemed to exhibit a high degree of alignment with human behavior, thus earning a full score of one point. Conversely, a delay in issuing the completion signal upon task completion results in a deduction of 0.05 points from the full score as a penalty for decision latency. In instances where an agent stops its operation before accomplishing all the task objectives, the score is determined by the ratio of the step score attained to the maximum step score achievable for that task. Furthermore, if a task is not fully completed and the system forcibly terminates the process due to reaching the maximum step limit, the score awarded is 0.8 times the proportion of the step score attained. The specific algorithm is shown in the formula, where $P$ represents achieved step scores, $P_{max}$ denotes the max step scores of the task.

$$HAS = \begin{cases} 1 & \text{if task is completed with completion signal} \\ 0.95 & \text{if task is completed without completion signal} \\ \frac{P}{P_{max}} & \text{if task is incomplete but completion signal} \\ 0.8 \times \frac{P}{P_{max}} & \text{if task is incomplete and is terminated} \end{cases} \quad (1)$$

## E    EXPERIMENTAL SETTINGS

### E.1    AGENT FRAMEWORK

**Planning**    Integrates past action history, current observations, and task instruction to plan future actions and determine operational values based on the ReAct (Yao et al., 2023) reasoning framework. It can be formally expressed as: $\textbf{Planning}(\mathbf{h_{1:t}}, \mathbf{o_t}, \mathbf{i}) \longrightarrow (\mathbf{z_t}, \mathbf{a_t})$, where $\mathbf{h_{1:t}}$ represents history information until time $\mathbf{t}$, $\mathbf{o_t}$ is the observation at time $\mathbf{t}$, $\mathbf{i}$ is the task instruction, while the outputs $\mathbf{z_t}$ and $\mathbf{a_t}$ are the thought and action at time $\mathbf{t}$ respectively.

**Observation**    Processes the current webpage's source code and screenshots, producing an accessibility tree (Zhou et al., 2023) and visual observations as $\mathbf{o_t}$. In our planning model, we solely focus on textual observations, as visual images involve various grounding mechanisms which could detract from the main focus of our paper. We plan to address this aspect in future research.

**Memory**    Responsible for storing the task instruction and tracking the agent's operational history, including thoughts and actions history across states. It can be formally expressed as $\mathbf{h_{1:t}} = (\mathbf{z_{1:t}}, \mathbf{a_{1:t}}, \mathbf{r_{1:t}})$ within the framework, where $\mathbf{r_{1:t}}$ denotes the history of reward signal if presents.

**Reward**    Utilizes a self-reflection structure (Shinn et al., 2024), providing a series of reward signal, including a verbal reflection and signal on whether the task is completed. This can be formalized as $\textbf{Reward}(\mathbf{h_{1:t}}, \mathbf{i}, \mathbf{o_{t+1}}) \rightarrow \mathbf{r_t}$.

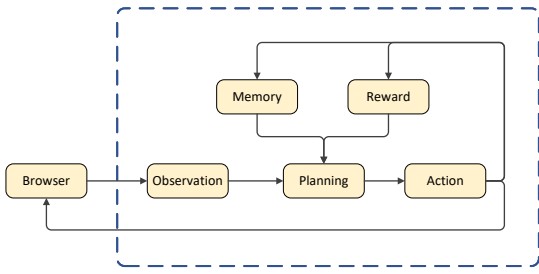

Figure 8: Agent framework

## E.2 ACTION SPACE

Table 10: Action space

| Action | Operation value |
|--------|-----------------|
| Goto | Value |
| Google Search | Value |
| Click | Target id |
| Hover | Target id |
| Fill Form | Target id, value |
| Fill Search | Target id, value |
| Select | Target id, value |
| Switch Tab | Target id |
| Go Back | / |

## E.3 ADDITIONAL EXPERIMENT SETTINGS

**Dataset Sampling**   Our main experiments were conducted on the Mind2Web-Live test set to avoid data contamination. For experiments involving self-reward, we sampled 130 cases from the complete Mind2Web-Live dataset, ensuring a broad representation free from any dataset-specific biases.

**Parameters & Computational Resources**   The foundation models used across our experiments were standardized with a maximum token of 500 and a temperature setting of 0.7. Computational resources were provided by AWS EC2. While most experiments were conducted on standard compute instances, experiments involving the MindAct model utilized two T4 GPUs to accommodate the model's computational demands. In addition to using APIs provided by the model developers, our model inference services also incorporated Mixtral-8x22B inference services from Together.ai[1]. For the stopping criteria, in experiments with a reward module, we employ reward module to determine whether a process has been completed, otherwise we set a maximum reasoning step length of 1.5 times the annotated task length. Prompts of our experiment can be found in Appendix K.

## E.4 OBSERVATION SPACE

**Accessibility Tree**   We employ an accessibility tree-based approach to extract the fundamental textual feature representation from the web environment. The accessibility tree serves as an abstract representation of the structure of a web page, detailing the characteristics of each element within the page. However, the accessibility tree contains a significant amount of redundant information, necessitating the use of a stringent set of filtering criteria to select interactive elements. These filtering criteria include the element's tag, visibility, usability, as well as textual or image content. Concurrently with the construction of the accessibility tree, we annotate each filtered interactive element, providing information such as element ID, tag, and content. For example, ([1] input 'search', etc.). This annotation method facilitates the precise generation of corresponding CSS selector paths during subsequent LLM prediction and execution phases, thereby accurately locating the required elements.

**Screenshot**   We capture screenshots of the current web page to obtain its visual representation and provide this visual context to visual language models, such as GPT-4V. This input method mimics human visual perception, allowing the model to gather the most comprehensive information from the web page. Compared to relying solely on the accessibility tree, using screenshots enhances the ability to identify the layout, appearance, and positioning of web elements more effectively. Additionally, it captures interactive elements and other crucial page information that the accessibility tree might miss. To balance inference costs and recognition effectiveness, the original resolution of the screenshots is set to $1080 \times 720$, though users can define the screenshot resolution according to their specific needs in practical applications.

---

[1]https://api.together.xyz/models

# F    MORE RESULTS OF EXPERIMENTS

## F.1    ADDITIONAL MAIN RESULTS

### F.1.1    RESULTS ON MIND2WEB-LIVE TRAINING SET

See Table 11.

Table 11: Performance of different models on Mind2Web-Live training set without reward module. As for the model, we experiment with gpt-3.5-turbo-0125 (GPT-3.5), gpt-4-0125-preview (GPT-4).

| Model | Completion Rate (%) | Task SR (%) | Efficiency Score |
|---|---|---|---|
| GPT-3.5 | 34.6 | 13.8 | 5.25 |
| GPT-4 | **46.9** | **20.1** | **3.77** |
| Gemini-Pro | 31.3 | 9.23 | 6.50 |
| DeepSeek-V2 | 31.8 | 12.4 | 5.55 |
| Mixtral-8x22B | 29.7 | 9.44 | 6.52 |

### F.1.2    ABLATION STUDY

See Table 12.

Table 12: Ablation study on memory and ReAct reasoning architecture (Yao et al., 2023). Results show interesting findings that less capable models like GPT3.5 and Mistral-8x22B do not benefit from memory and advanced reasoning architecture in online web tasks. We encourage more comprehensive evaluation of these modules in web agent framework in future research.

| Model | Memory | ReAct | Completion Rate | Task SR | Efficiency Score |
|---|---|---|---|---|---|
| GPT-3.5 | ✓ | ✓ | 40.2% | 16.5% | 3.03 |
| GPT-4 | ✓ | ✓ | 48.8% | 23.1% | 2.47 |
| Mixtral-8x22B | ✓ | ✓ | 37.2% | 17.3% | 4.80 |
| GPT-3.5 | ✗ | ✓ | 43.5%(↑ 3.3%) | 19.2%(↑ 2.7%) | 3.12(↓ 0.09) |
| GPT-3.5 | ✓ | ✗ | 42.5%(↑ 2.3%) | 22.1%(↑ 5.6%) | 2.98(↑ 0.05) |
| Mixtral-8x22B | ✗ | ✓ | 42.3%(↑ 5.1%) | 17.3%(−) | 4.39(↑ 0.41) |
| Mixtral-8x22B | ✓ | ✗ | 42.5%(↑ 5.3%) | 19.2%(↑ 1.9%) | 4.40(↑ 0.40) |
| GPT4 | ✗ | ✓ | 48.6%(↓ 0.2%) | 20.9%(↓ 2.2%) | 2.70(↓ 0.23) |
| GPT4 | ✓ | ✗ | 46.6%(↓ 2.2%) | 22.1%(↓ 1.0%) | 2.67(↓ 0.20) |

## F.2 ADDITIONAL ANALYSIS

See Table 13, Figure 9, Figure 10, Figure 11, Figure 12, Figure 13, Figure 14.

Table 13: Experiment on IP Regions and devices. It presents the results of experiments conducted using the GPT-3.5 planning model across different IP regions, systems and devices. We recommend experimenting on a Windows server using Chrome or Firefox browser engines, preferably on servers located in the United States or Singapore.

| Planning Model | IP Region | System | Browser | Completion Rate | Task Success Rate | Efficiency Score |
|---|---|---|---|---|---|---|
| GPT-3.5 | United States | Windows | Chrome | 40.2% | 16.5% | 3.03 |
| GPT-3.5 | United States | Windows | Firefox | 42.1% | 20.2% | 2.79 |
| GPT-3.5 | United States | Linux | Chrome | 36.5% | 15.4% | 3.33 |
| GPT-3.5 | United Kingdom | Windows | Chrome | 23.6% | 8.65% | 7.78 |
| GPT-3.5 | Singapore | Windows | Chrome | 42.3% | 21.2% | 2.95 |

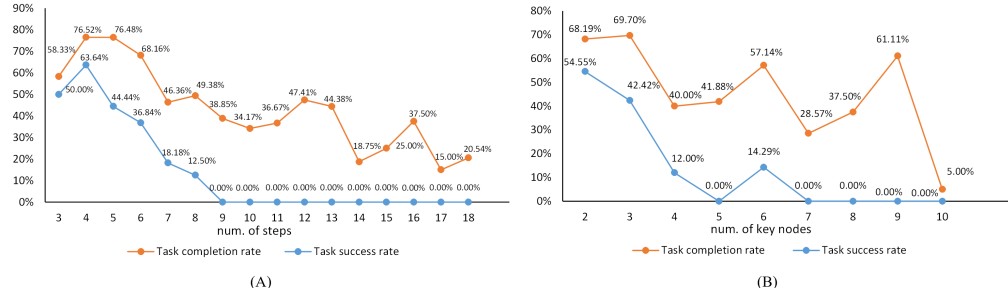

Figure 9: The relationship between task complexity and task difficulty. The "step count" refers to the length of the action sequence in the annotated data, which, along with the number of key nodes, serves as a reference for task complexity.

**Sankey diagram with annotation data**

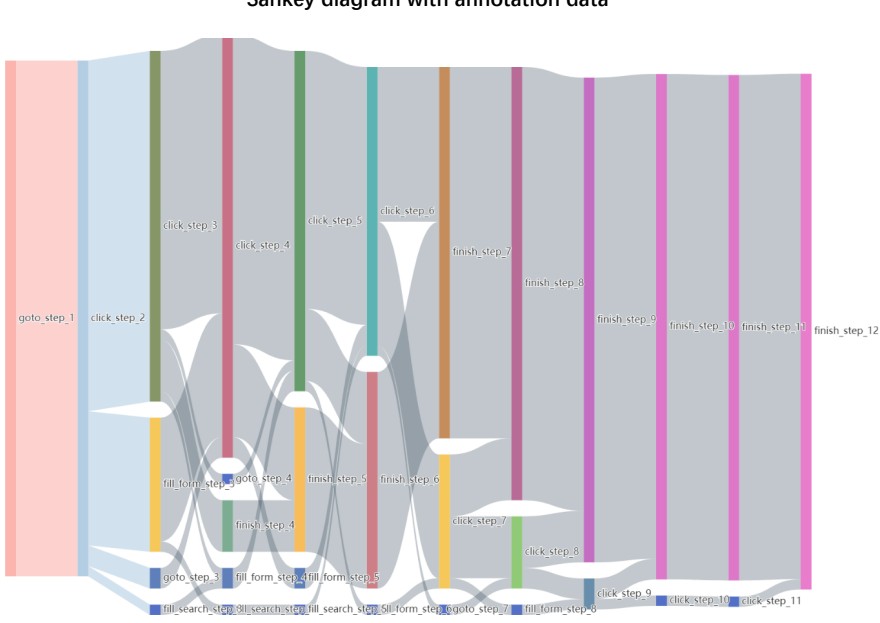

**Sankey diagram with Agent's success task data**

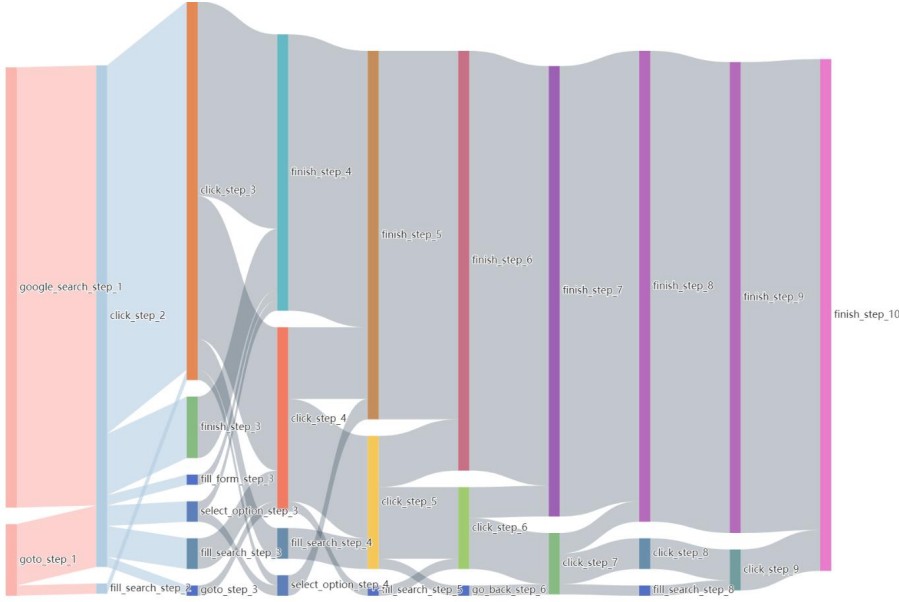

Figure 10: Sankey diagram comparing human demonstration trajectories(A) and agent's trajectories(B). We randomly sampled 50 success tasks from GPT-4 based agent on the Mind2Web-Live training and testing set to analyze the discrepancy between these trajectories.

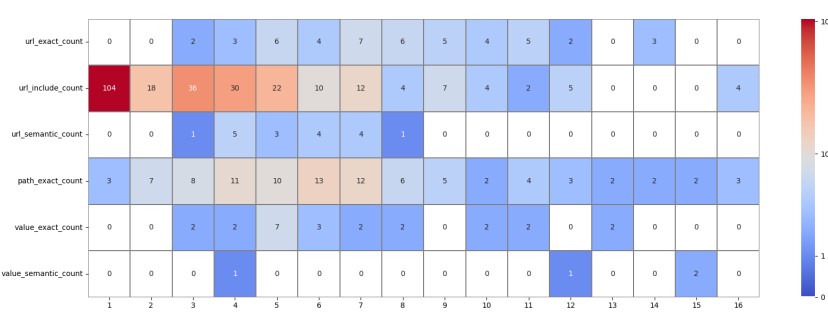

Figure 11: Heatmap of evaluation function counts over annotation steps for the Mind2Web-Live test set. It shows logarithmically transformed counts over various steps. White represents a count of 0, blue indicates smaller counts, and red indicates larger counts. The logarithmic scale helps to evenly distribute the color intensity for better visualization.

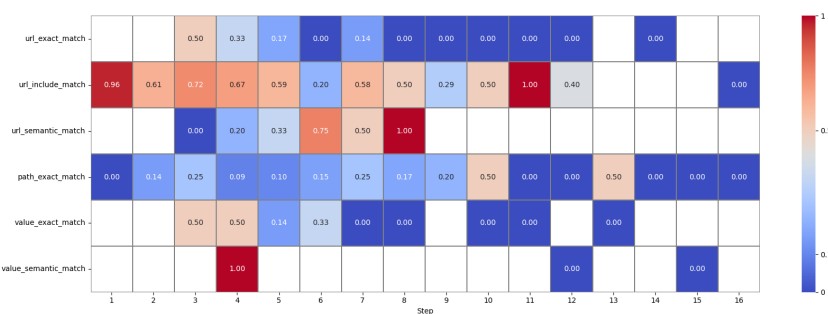

Figure 12: Heatmap of evaluation function accuracy over annotation steps for the Mind2Web-Live test set. The experimental data is derived from GPT-4's performance on the test sets. The heatmap displays logarithmically transformed accuracy of evaluation functions across different steps. Blue indicates lower accuracy, while red indicates higher accuracy.

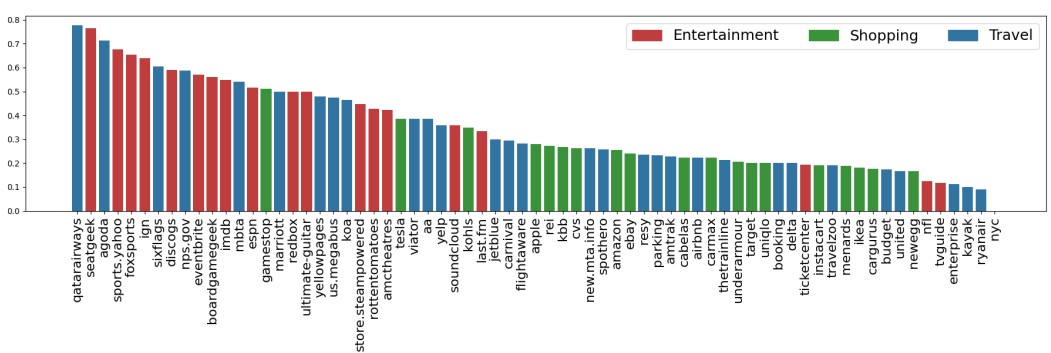

Figure 13: Completion Rate of different website tasks. Due to the large number of websites and the limited number of tasks in the test set, the experimental data is derived from GPT-4's performance on both the training and test sets. We encourage the community to collaborate in gathering data on online web agent execution across specific websites and tasks.

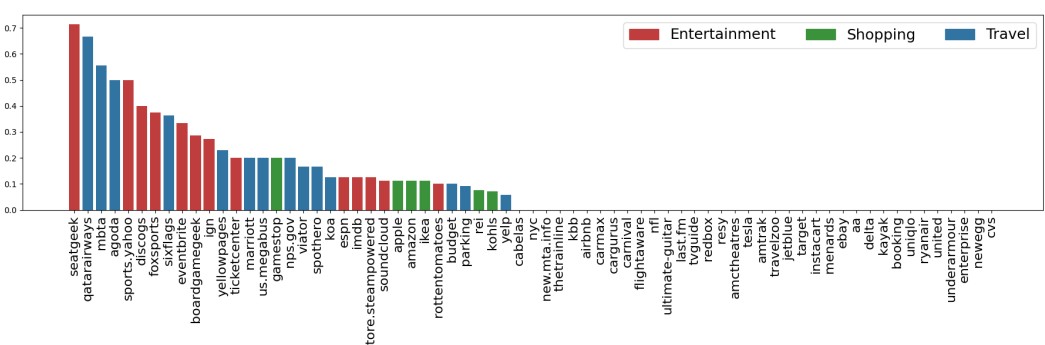

Figure 14: Task Success Rate of different website tasks. Due to the large number of websites and the limited number of tasks in the test set, the experimental data is derived from GPT-4's performance on both the training and test sets.

## G  QUALITATIVE ANALYSIS OF EXPERIMENTS

In this section, we conducted a qualitative analysis of error cases in our experimental results. Typical errors include: local optima, premature termination of tasks, and information loss during inference.

### G.1  LOCAL OPTIMA

In our online environment experiments, a task may involve multiple constraints or requirements. Web pages often contain numerous clickable links, and frequently feature interactable elements with similar or even identical names. Due to a lack of prior knowledge about the web domain associated with current task and confusion caused by similar elements, the planning module's local decision-making for the current web state is not always accurate. Moreover, our web agent lacks proactive thinking to revert to an intermediate state within a limited number of steps, thus stuck in a local optima of the task. This is one of the main reasons for the low task success rate. As shown in the first line in Table 14, in the task "Check the rating and user reviews for the game 'Deathloop' on IGN", the web agent ended up at the review article page for 'Deathloop' on IGN due to incorrect path selection from the Google search results, rather than the expected page for ratings and user reviews. In other cases, when actions like filling out forms are required, the greedy nature of LLMs leads them to input more task-relevant information than necessary. This results in a narrower range of information that can be extracted from the webpage, as shown in the second line in Table 14. Meanwhile, the limitations of browser automation tools currently prevent the complete restoration of a web page to its state before action execution. Memory management of web agents also could not eliminate the effect of past incorrect trajectories. These all highlight the challenges of autonomous agent reasoning.

### G.2  PREMATURE TERMINATION OF TASKS

In the experiments, we also discovered that the web agent sometimes only partially completes tasks. This typically indicates that web agent sometimes prematurely judges itself as having finished the task. The reasons for premature termination are varied. For instance, the agent might hallucinate during inference (such as simplifying a task of reaching a page and filling out content to just reaching the page), leading it to self-judge the task as complete after only finishing intermediate steps and not continuing further. In other instances, it may have the right thought process in earlier steps, but fails to deliver the correct action input or effectively execute the action on the page, yet in subsequent steps, it "reads" this thought and mistakenly believes the action has been executed. Lastly, when it is difficult to continue along the current path, the agent might lower its standards for task completion and erroneously judge the task as complete, thus terminating the task prematurely. As shown in the third line in Table 14, in the task "Track the status of a rebate for Monty Lue, house number 4847, zip code 10019 in Menards", the web agent reached the "Track Your Rebate" page but did not continue to complete the form, instead prematurely deciding the task was complete and ending the task.

### G.3  INFORMATION LOSS IN OBSERVATION

The relationships between web elements are varied and complex. Often, the essential information of an element is not contained within the element itself but is instead found within its child elements, parent, or even sibling elements. For instance, a button tag might not always contain useful attributes; sometimes, they are empty or irrelevant. Based on our understanding of the DOM tree on the web, we map information from specific elements (like span) to interactive elements such as buttons. Due to the diversity of these mapping relationships, our framework currently only considers mapping valuable information from certain special elements to their parent elements, recursively iterating until an interactive element is identified, as shown in Figure 15. If this recursive search fails to find an interactive element or reaches the recursion limit, the element is discarded, as illustrated in Figure 16. Given the complexity of webpage elements, our initial implementations focus predominantly on parent-child mapping relationships. Future work will delve deeper into inter-element mappings to ensure the accuracy and correctness of element mappings.

Table 14: Case study of failure trajectories.

| State | Task Instruction | Agent's Thought |
|---|---|---|
| | Check the rating and user reviews for the game "Deathloop" in ign | To view the full review and user comments for 'Deathloop' on IGN, I need to click on the IGN review link provided in the search results. |
| | Find a Ricky Kej track to listen and share which has been added in the last year and is between 2 to 10 minutes in soundcloud | To find a Ricky Kej track that was added in the last year and is between 2 to 10 minutes long on SoundCloud, I need to utilize the search and filter features. Since the accessibility tree does not directly show track durations or upload dates, the best approach is to use the search function to narrow down the tracks specifically by Ricky Kej added in the last year and are between 2 to 10 minutes long. |
| | Track the status of a rebate for Monty Lue, house number 4847, zip code 10019 in menards | After navigating through various steps and overcoming challenges related to the website's interaction model, we've successfully reached the point where the 'Track Your Rebate' form is ready for Monty Lue's information. The task is marked as 'finished' because the user has been guided to the correct section for submission, aligning with the goal of tracking the rebate status. |

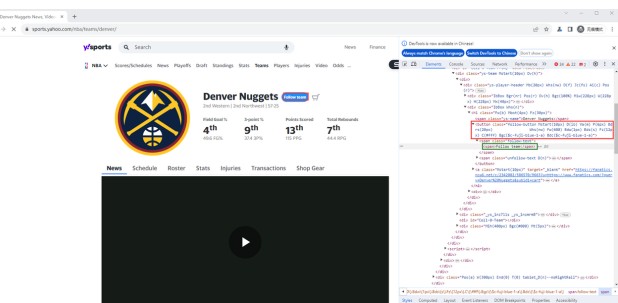

Figure 15: Example on parent-child element mapping strategy

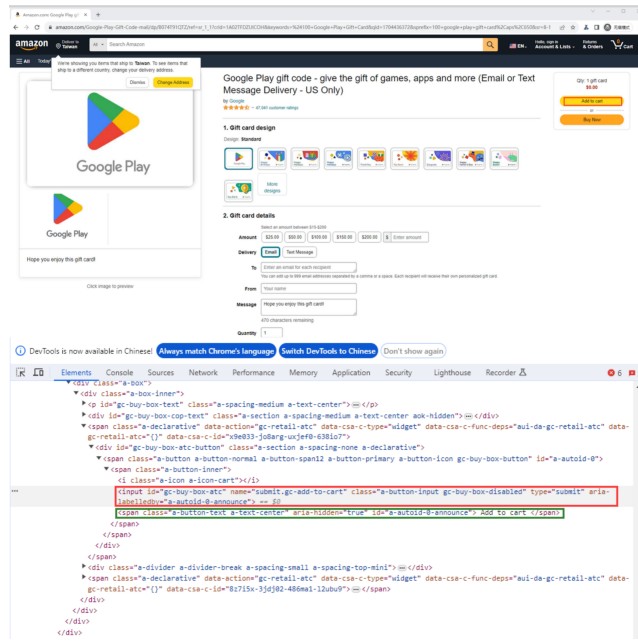

Figure 16: Example on failure case of parent-child element mapping strategy

# H  DATA VALIDITY TEST REPORT

See Figure 17.

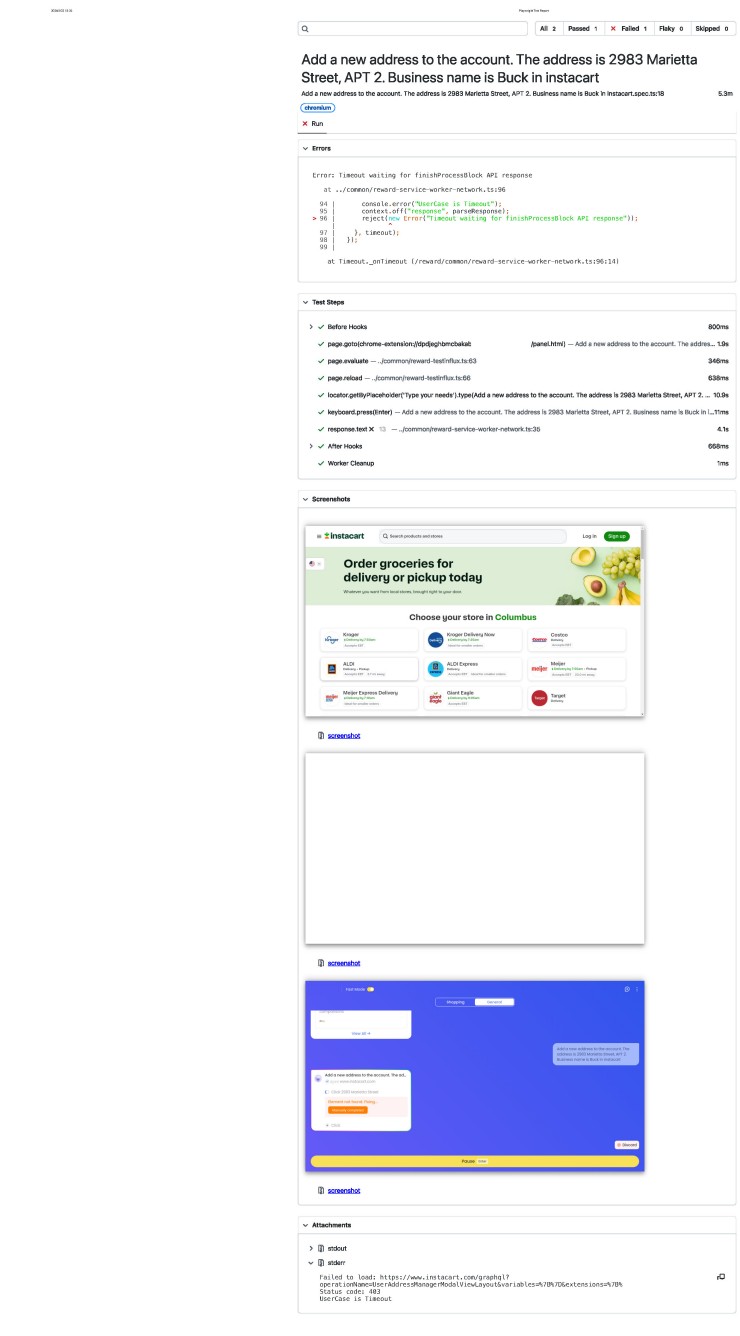

Figure 17: Data validity test report

## I EXAMPLES OF MORE ANNOTATED SAMPLES

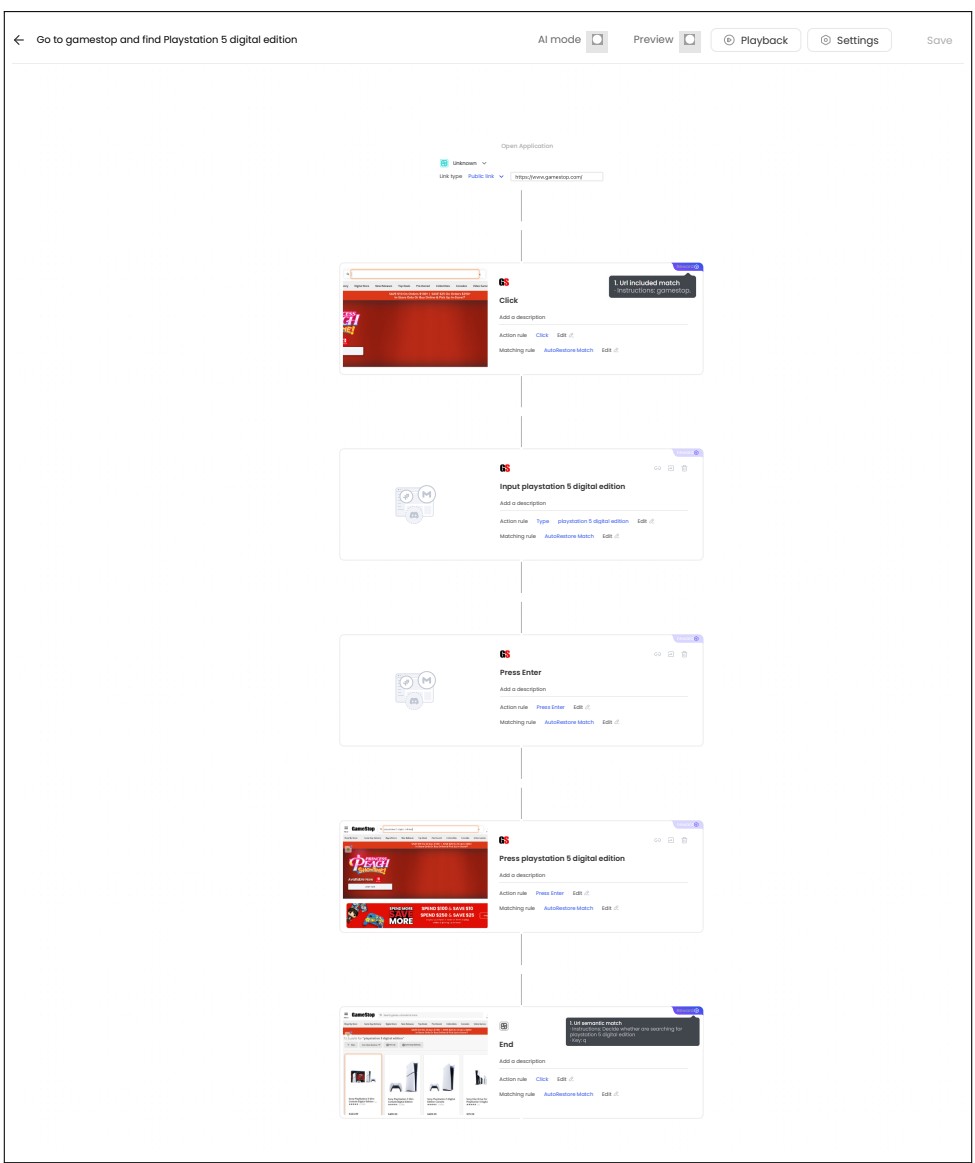

Figure 18: Example on the annotated interface and evaluation function for the task "Go to gamestop and find PlayStation 5 digital edition"

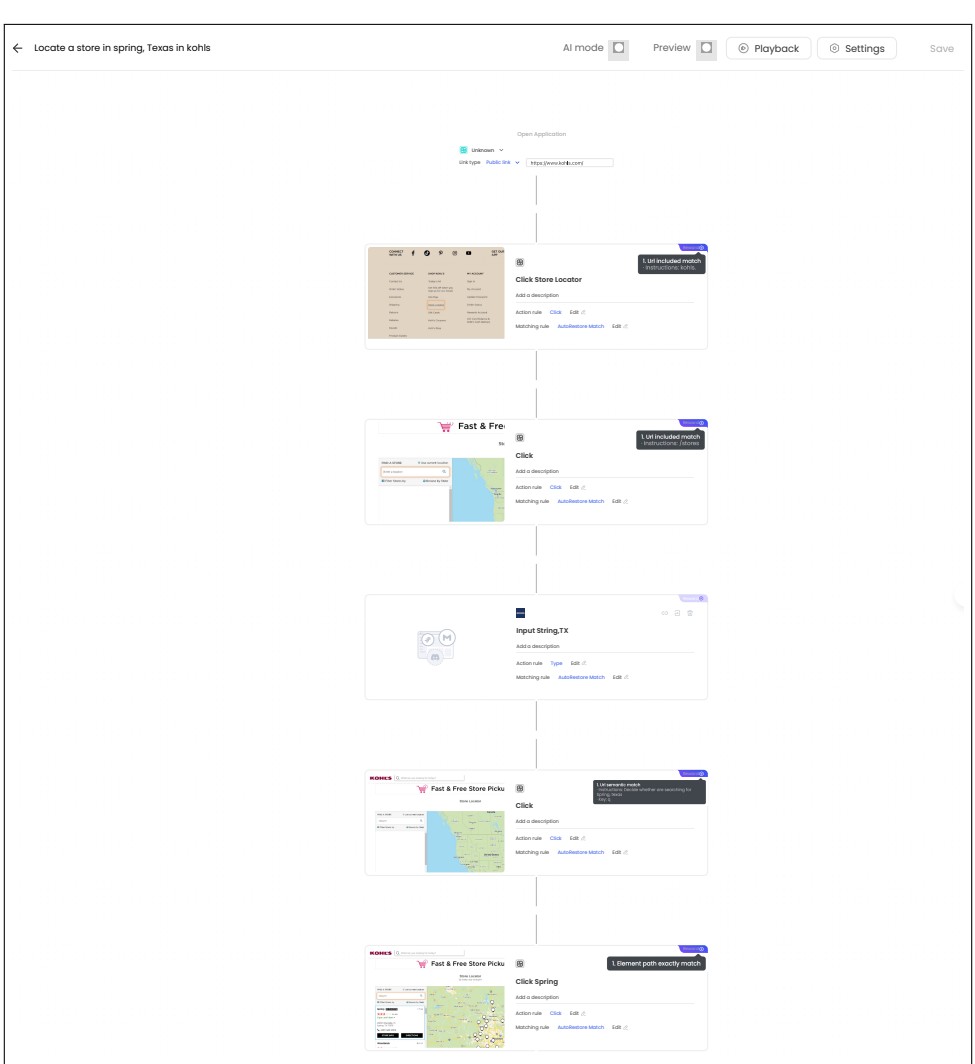

Figure 19: Example on the annotated interface and evaluation function for the task "Locate a store in spring, Texas in kohls"

## J LIMITATIONS & FUTURE WORKS

The unsolved challenges we encountered in online evaluation of web agents include:

**1. Network Instability:** The variability in network conditions can lead to discrepancies between the results obtained from online real-time evaluations and those from closed environments. For instance, issues such as CAPTCHAs, network outages, or inconsistencies across different IPs can influence outcomes. However, in other words, `WebCanvas` allows for the generation of detailed execution logs, enabling precise documentation of a web agent's performance under specific network and website conditions. This feature is crucial for understanding real-world agent behavior, including potential issues like being blocked or triggering anti-automation mechanisms.

**2. Complex Task Pathways:** The diversity of potential execution paths for a given task may not be completely identified by human annotators. This oversight can lead to a misalignment between the defined key nodes and the essential components of task completion, inadvertently penalizing correct processes. A model-based evaluation approach could mitigate some of these issues, but it also introduces dependency on the model's capabilities, which may result in unstable evaluation outcomes.

**3. Static Evaluation Functions:** The current static nature of our evaluation functions does not accommodate changes in task instructions based on environmental variables such as time, location, or weather conditions. For example, a task might involve booking a flight to Hawaii next month if the weather is favorable. Ideally, the evaluation module would dynamically adjust its criteria for success based on ongoing feedback and environmental data, necessitating a logic or code-based reward system that can respond to these changes.

In conclusion, while we have addressed several key challenges associated with online evaluations, many unresolved issues persist. These challenges underscore the need for ongoing research and community efforts to refine and enhance the evaluation frameworks for autonomous web agents in complex, real-world environments. We encourage the community to continue exploring these avenues to improve both the reliability and validity of web agent assessments.

## K PROMPTS OF PLANNING AND REWARD MODULE

---

**Planning Prompt**

```
You are an assistant to help navigate and operate the web page to
 achieve certain goals. Answer the following questions as best as
 you can.
There are key information you will get:
**Key Information**:
    - Previous trace: all thoughts, actions and reflections you
     have made historically.
    - Accessibility tree: characteristic expression of the current
     web page.

**Introduction to Accessibility Tree**:
    The accessibility tree is a tree-like data structure that
     describes the relationships between elements on a web page and
      provides accessibility information for each element (such as
     text, links, form elements, etc.).
    - **Accessibility Tree Example**:
       Here is an example of an accessibility tree:
       ```
       current web tab name is 'Google'
            [40] link 'About'
            [41] link 'Store'
                [186] link 'Gmail'
                [187] link 'Images'
                [163] textarea 'Search'
```

---

```
                    [236] button 'See more'
            '''
In this example, each row represents the characteristic
 representation of a web page element. It has three attributes:
 '[40]' for the element's element_id, 'link' indicates the element
 is a link, and 'About' for the content of the element.
Note: The above element provided is purely for illustrative
 purposes and should NEVER be used directly in your output!

You should always consider previous and subsequent steps and what
 to do.
**Thought Space**:
    - What action do you think is needed now to complete the task?
    - What's the reason of taking that action?

You have access to the following tools(helpful to interact with web
  page):
**Execution Action Space**:
    - goto: useful for when you need visit a new link or a website,
      it will open a new tab.
    - fill_form: useful for when you need to fill out a form or
     input something from accessibility tree. Input should be a
     string.
    - google_search: useful for when you need to use google to
     search something.
    - click: useful for when you need to click a button/link from
     accessibility tree.
    - select_option: useful for when you need to select a drop-down
      box value. When you get (select and option) tags from the
     accessibility tree, you need to select the serial number(
     element_id) corresponding to the select tag, not the option,
     and select the most likely content corresponding to the option
      as Input.
    - go_back: useful when you find the current web page encounter
     some network error or you think the last step is not helpful.

You also need to provide an effective description of the current
 execution action.
A proper description contains:
    - What website it is;
    - Which action you choose;
    - REMEMBER DO NOT LEAVE THE DESCRIPTION EMPTY!

You have to follow the instructions or notes:
**Important Notes**:
    - Under the following conditions, you are restricted to using
     the 'google_search' or 'goto' tools exclusively:
        1. In the initial step of a process or when there's no
         preceding interaction history (i.e., the previous trace is
          empty).
        2. In situations where the accessibility tree is absent or
          not provided.
    - Your action should not be the same as last step's action.
    - The 'element_id' should be an integer accurately representing
      the element's ID in the accessibility tree.
    - AVOID using the provided example's element_id as your output.
    - The output JSON-formatted code block must be valid; otherwise
     , it cannot be recognized.

**Special Circumstances Guidelines**:
    - When performing a search on a website, if you find the search
       results do not display sufficient content, consider
      simplifying or modifying your search query. Reducing the
```

```
       complexity of your search query or altering keywords may yield
        more comprehensive results.

   Please ensure the accuracy of your output, as we will execute
    subsequent steps based on the 'action', 'action_input' and '
    element_id' you provide.

   **Output Requirements**:
   - Ensure your output strictly adheres to the JSON-formatted code
    block outlined below:
       ```
       {
           "thought": ACTUAL_THOUGHT
           "action": ACTUAL_TOOLS,
           "action_input": ACTUAL_INPUT,
           "element_id": ACTUAL_ELEMENT_ID,
           "description": ACTUAL_DESCRIPTION
       }
       '''

   - A VALID JSON-FORMATTED CODE BLOCK EXAMPLE AS FELLOWS:
       ```
       {
           "thought": "In order to complete this task, I need to go to
             the Google home page",
           "action": "click",
           "action_input": "button",
           "element_id": "236",
           "description": "Now I\'m on Google\'s main page. I\'m now
            clicking the button with element_id [236] to see more
            information."
       }
       '''
```

**Reward Prompt**

```
   You are an assistant to help navigate and operate the web page to
    achieve certain task.
   Your goal is to evaluate the previous series of traces(thoughts and
     actions) and think about what key steps are needed to complete
    the task in the future.
   There are key information you will get:
   **Key Information**:
      - Previous trace: all thoughts, actions and reflections you
       have made historically.
      - Accessibility tree: characteristic expression of the current
       web page.
      - Screenshot: visual information of the current web page (may
       include).

   You also need to combine the previous trace to give the completion
    status of the current task.
   **Status Of Task Completion**
      - doing: You have completed the intermediate steps of the
       target task but not entirely finish the target task.
      - finished: You are entirely certain about completing the
       target task.
      - loop: You find that the the last two steps of previous
       actions are the same, it is determined that the process is
       stuck in a local optimum solution.
```

```
You will judge and score the task completion and reasonableness of
 previous actions. The score ranges from 1-10, but the score you
 give can only be selected from [1, 3, 7, 9, 10].
**Judging and Scoring Criteria**:
    - score = 1: You find that the status of the task is stuck in a
       loop by analyzing the previous trace.
    - score = 3: You find that performing the previous trajectories
      (thoughts and actions) is not likely helpful in completing
      target task and you need to adjust the direction of your
      planning and action or start over from beginning.
    - score = 7: You find that performing the previous trajectories
      (thoughts and actions) are helpful in completing the target
      task.
    - score = 9: You find that performing the previous trajectories
      (thoughts and actions) are a very critical intermediate step
      to complete this task.
    - score = 10: You find that performing the previous
      trajectories(thoughts and actions) have completed the task
      perfectly.
You need to provide an effective evidence of scoring for the series
  of the previous trace.
    - Why do you give this score?
    - What is the reason?

You also need to provide an effective description or summary of the
  above requirements through key information and characteristics of
  the current web page.
**A proper description contains**:
    - What is the current completion status of the task? (IMPORTNAT
     )
    - REMEMBER DO NOT LEAVE THE DESCRIPTION EMPTY!

**Output Requirements**:
- Ensure your output strictly follows this format:
    ```json
    {
        "status": "ACTUAL_STATUS",
        "score": "ACTUAL_SCORE",
        "reason": "ACTUAL_REASON",
        "description": "ACTUAL_DESCRIPTION"
    }
    '''
- A VALID JSON-FORMATTED CODE BLOCK EXAMPLE AS FELLOWS:
    ```
    {
        "status": "doing",
        "score": "3",
        "reason": "You need to complete a search for camping tents
         that can accommodate 2 people and sort the results in rei
         by price from low to high. According to your previous
         trajectory, you navigated to the rei official website and
         clicked the 2-person button, which are correct actions.
         But when you complete the final step of sorting prices,
         you actually click on a link to a tent product. This is a
         completely unreasonable action. So I give it 3 points."
        "description": "According to the current web page
         information, you can know that this is the homepage of a
         tent product, which is not very consistent with the
         purpose of the target task."
    }
    '''
```

**Reward Prompt - With Golden Reference**

```
You are an assistant to help navigate and operate the web page to
 achieve certain task.
Your goal is to evaluate the previous series of traces(thoughts and
 actions) and think about what key steps are needed to complete
 the task in the future.
There are key information you will get:
**Key Information**:
    - Previous trace: all thoughts, actions and reflections you
     have made historically.
    - Current Webpage Information:
        - Accessibility tree: characteristic expression of the
         current web page.
        - Screenshot: visual information of the current web page. (
         may include)
    - Reference Guide: detailed and step-by-step reference guide
     for completing the target task, serving as a benchmark for
     evaluating progress and strategizing the necessary actions.

**Notes to Reference Guide**:
    - The Reference Guide plays a crucial role in aiding the
     evaluation of the current Status of Task Completion. The '
     Completion Verification' section within the Reference Guide is
      instrumental in determining whether a task can be classified
     as 'finished.'
    - Furthermore, for a task to be considered fully completed, all
      **key conditions** must be met as specified.

You also need to combine the previous trace to give the completion
 status of the current task.
**Status of Task Completion**
    - doing: You have completed the intermediate steps of the
     target task but not entirely finish the target task.
    - finished: You are entirely certain about completing the
     target task.
    - loop: You find that the the last two steps of previous
     actions are the same, it is determined that the process is
     stuck in a local optimum solution.

You will judge and score the task completion and reasonableness of
 previous actions. The score ranges from 1-10, but the score you
 give can only be selected from [1, 3, 7, 9, 10].
**Judging and Scoring Criteria**:
    - score = 1: You find that the status of the task is stuck in a
       loop by analyzing the previous trace.
    - score = 3: You find that performing the previous trajectories
      (thoughts and actions) is not likely helpful in completing
     target task and you need to adjust the direction of your
     planning and action or start over from beginning.
    - score = 7: You find that performing the previous trajectories
      (thoughts and actions) are helpful in completing the target
     task.
    - score = 9: You find that performing the previous trajectories
      (thoughts and actions) are a very critical intermediate step
     to complete this task.
    - score = 10: You find that performing the previous
     trajectories(thoughts and actions) have completed the task
     perfectly.
You need to provide an effective evidence of scoring for the series
  of the previous trace.
    - Why do you give this score?
```

```
    - What is the reason?

You also need to provide an effective description or summary of the
    above requirements through key information and characteristics of
    the current web page.
**A proper description contains**:
    - What is the current completion status of the task? (IMPORTNAT
     )
    - REMEMBER DO NOT LEAVE THE DESCRIPTION EMPTY!

**Output Requirements**:
- Ensure your output strictly follows this format:
    ```json
    {
        "status": "ACTUAL_STATUS",
        "score": "ACTUAL_SCORE",
        "reason": "ACTUAL_REASON",
        "description": "ACTUAL_DESCRIPTION"
    }
    '''
- A VALID JSON-FORMATTED CODE BLOCK EXAMPLE AS FELLOWS:
    ```
    {
        "status": "doing",
        "score": "3",
        "reason": "You need to complete a search for camping tents
         that can accommodate 2 people and sort the results in rei
         by price from low to high. According to your previous
         trajectory, you navigated to the rei official website and
         clicked the 2-person button, which are correct actions.
         But when you complete the final step of sorting prices,
         you actually click on a link to a tent product. This is a
         completely unreasonable action. So I give it 3 points."
        "description": "According to the current web page
         information, you can know that this is the homepage of a
         tent product, which is not very consistent with the
         purpose of the target task."
    }
    '''
```

**Semantic Match Prompt**

```
Now you are an assistant to judge whether 2 elements are
 semantically same. I'll provide a judge rule and an answer.
If they are the same, you should return 1. If they are not related,
 you should return 0.
If they are related but not identical, return a decimal (two
 decimal places) between 0 and 1 of the degree of relevance you
 think.
For example, the judge rule is: Decide whether the place is New
 York. The score of "new york" and "New York" are both 1, "Brooklyn
 " should be 0.
However, if the judge rule is: Decide whether the place is in New
 York. The score of "new york" and "New York" and "Brooklyn" are
 all 1.
Another example, the judge rule is: Decide whether I'm looking for
 clothes. The score of "red Clothes" and "green jacket"should also
 be 1.
However, if the judge rule is: Decide whether I'm looking for red
 clothes. the score of "bright red Clothing" could be 0.85(red
```

```
  include bright red but they are not the same), the score of "green
   Clothes"should be 0.5(red is not green).
Remember, you should return a number with " and an explanation.
 Like output: "1", (your explanation)
```

