# OpenReview forum: "WebCanvas: Benchmarking Web Agents in Online Environments"
_ICLR.cc/2025/Conference — ICLR 2025 Conference Withdrawn Submission_

### Official Review · Reviewer_m7E8 · 2024-11-03

**Soundness:** 2
**Presentation:** 3
**Contribution:** 2
**Rating:** 5
**Confidence:** 3

**Summary:**

This paper introduces a novel framework for assessing web agents in dynamic online environments. In contrast to conventional benchmarks that focus on static web conditions, WebCanvas proposes a novel key-node-based evaluation metric, an enhanced dataset named Mind2Web-Live, and efficient annotation tools. Additionally, the authors demonstrate their best-performing agent in the Mind2Web-Live dataset and provide the analysis of the performance discrepancies.

**Strengths:**

+ This paper is mostly well-written and easy to follow.
+ The paper is technically sound with most claims supported sufficiently by experimental results.
+ The proposed evaluation metrics and datasets seem novel.

**Weaknesses:**

- The problem formulation is incomplete in Section 2. The authors should bring some contents in Section E.1 back to the main paper. Additionally, the final objective function is missing in Section 2 as well.
- It is a bit odd that “include match” and “semantic match” share the same evaluation targets for step score. Not sure if it is better to introduce additional aspects to distinguish them.
- Some parts of the presentation could be improved, e.g., in Line 136, the notation of action history a_{1}^{t-1} is not clear. It is better to use a_{1:t-1} to represent history following POMDP literature.

**Questions:**

- The problem formulation is incomplete in Section 2.
- The authors might consider bringing some contents in Section E.1 back to the main paper.
- The final objective function seems to be missing in Section 2.
- “include match” and “semantic match” share the same evaluation targets for step score. Consider introducing additional aspects to distinguish them.
- Some parts of the presentation could be improved, e.g., in Line 136, the notation of action history a_{1}^{t-1} is not clear.
- It is better to use a_{1:t-1} to represent history following POMDP literature.

---

> ### Author Response · Authors · 2024-11-18
> **Authors’ Response (Part 1 of 2)**
>
> Thank you for taking the time to review our paper, and we truly appreciate your recognition of the contributions of WebCanvas!
>
> We believe that all your concerns can be addressed during the discussion phase. Please refer to our responses below. If there are any remaining unclear points, we would be more than happy to further clarify them during the discussion.
>
> # 1. Clarification on Symbol Representation
>
> We sincerely thank you for pointing out the issue with our symbol representation. We fully accept this suggestion. The original notation $a_{1}^{t-1}$ may not be clear. We will revise the notation to $a_{1:t-1}$ in the relevant sections (L136, L837) of the manuscript.
>
> # 2. Response to Issues in Section 2 and Framework Description
>
> **On the problem formulation:** In the problem formulation, we referenced the formulation used in WebArena[1]-the well-recognized paper in web agents. We intentionally avoided mentioning the original POMDP formulation because it might confuse some readers in our setting. WebCanvas uses a Key Node-based Evaluation approach and does not involve a traditional "Final Objective Function." This is because WebCanvas focuses on progress evaluation during the task execution. A task is deemed successful if the agent completes all the step scores associated with the task's key nodes. This approach is fundamentally different from traditional benchmarks, which rely on either Reference-based or Outcome-based evaluations. Figure 3 in the paper illustrates the advantages of Key Node-based Evaluation, particularly in its ability to handle the dynamic and diverse nature of real-world web environments.
>
> **On Appendix E.1 content placement:** We agree that Appendix E.1 provides additional details about the experimental framework. However, we decided to include this content in the appendix for two primary reasons:
>
> - **Length considerations:** Due to the page limit, we prioritized the core contributions and innovations of our work. For readers familiar with prior work, the concise description in Section 5 is sufficient to understand our framework and experimental methodology. For readers less familiar with the background, Appendix E.1 offers a clear and comprehensive explanation.
>
> - **Avoiding potential misunderstanding:** Our agent framework is a baseline framework and does not introduce significant innovations compared to existing frameworks such as WebArena[1] or VisualWebArena[2]. By placing this content in the appendix, we aim to prevent any misinterpretation that our framework itself represents a major contribution. This paper is mainly innovating on introducing a define-and-evaluate scalable evaluation framework to connect web agents to live environment and we thoroughly jusify its effectiveness.
>
> > [1] WebArena: A Realistic Web Environment for Building Autonomous Agents, Zhou et al., 2023
> >
> > [2] VisualWebArena: Evaluating Multimodal Agents on Realistic Visual Web Tasks, Koh et al., 2024

---

> > ### Author Response · Authors · 2024-11-18
> > **Authors’ Response (Part 2 of 2)**
> >
> > # 3. Clarification on "Include Match" and "Semantic Match"
> >
> > Thank you for raising this question. However, we are not entirely certain what you mean by “share the same evaluation targets for step score.” Based on our understanding, your question might be asking, “Why can include match and semantic match both be used for the same evaluation target (e.g., URL)?” Below is our clarification:
> >
> > In the Mind2Web-Live dataset, each key node is explicitly assigned a single, unique Evaluation Function. Specifically, a key node’s evaluation target (e.g., URL) is assigned only one match type (e.g., include match or semantic match) and never both simultaneously. It is not possible for the same element to have both URL include match and URL semantic match as Evaluation Functions. During data annotation, the choice of match type depends on the semantic requirements of the task and the characteristics of the web element. Our annotation team underwent rigorous training and multiple rounds of trial annotation (detailed in Appendix A.2) to ensure consistency and accuracy in the selection process.
> >
> > - **Semantic Match:** This is used for content with diverse expressions. For example, when searching for “iPhone 16” on a shopping website, the results may include “iPhone 16” or “iphone16.” Such variations cannot be identified using simple substring matching and require semantic matching. Specially, if the evaluation target is url, semantic match function is often associated with a key in the url.
> >
> > - **Include/Exact Match:** These are applied to fixed or partially fixed parameters. For instance, in a URL containing a filter parameter for color (“color=black”), the parameter can be checked using either include match or exact match.
> >
> > By designing key nodes and annotating Evaluation Functions appropriately, we propose an evaluation approach that is more reflective of real-world environments. This approach effectively addresses challenges posed by dynamic web content and diverse task paths. Additionally, it distinguishes itself from traditional Reference-based or Outcome-based evaluations by offering a more comprehensive assessment of agent capabilities in web navigation tasks.
> >
> > # 4. Final Remarks
> >
> > WebCanvas is a step forward to advance the evaluation of real-world agents in live environments, which is crucial in web agents domain. By introducing the Key Node-based evaluation framework alongside the Mind2Web-Live dataset, our framework offers a scalable and in-progress evaluation in dynamic, live environments, with many existing methods struggling to solve effectively. Furthermore, we are actively working with the community to improve our data quality and refresh outdated datasets, ensuring our framework remains reliable and relevant over time. While our work can still be improved, we remain optimistic about its potential impact and believe it offers a fresh perspective and possibilities for advancing the field.
> >
> > If you have any further questions or concerns regarding the manuscript, please do not hesitate to comment! We are deeply passionate about this work and value your feedback and suggestions. We hope that our efforts to address the feedback, along with the contributions and potential impact of this work, might encourage you for a re-evaluation of this work. Thank you again for your thorough review.

---

> ### Author Response · Authors · 2024-11-30
> **Thank You and Looking Forward to the Discussion**
>
> Dear Reviewer m7E8, We wanted to say in advance our heartfelt thank you's for the time and effort you've put into both the reviews that have passed as well as the upcoming current discussion period. We know that you all have your own papers that you have to deal with during this busy time, and sincerely appreciate the time you've taken to spend on ours.
>
> We are so excited about this paper and its findings, so we are very much looking forward to the upcoming discussions with you! Please don't hesitate to ask us any questions big or small, and we are happy to provide any further clarifications.

---

### Official Review · Reviewer_BnDr · 2024-11-04

**Soundness:** 2
**Presentation:** 3
**Contribution:** 2
**Rating:** 5
**Confidence:** 3

**Summary:**

This paper presents a novel benchmark for web-based tasks designed for agent evaluation. The proposed benchmark introduces step-wise assessment, live sample testing, and a user-friendly community platform, facilitating the online evaluation of agents. The authors conduct experiments using several large language models (LLMs) on the proposed platform.

**Strengths:**

1. The paper contributes a significant new benchmark for web mining, which is expected to provide substantial value to the research community.

2. The benchmark incorporates several valuable features, including intermediate state evaluations, a user-friendly interface with plugin support, and access to live datasets.

3. The writing is clear and well-structured, with numerous case studies that aid in understanding the framework and its applications.

**Weaknesses:**

1. The experimental evaluation is limited to a comparison with Mind2Web. It would be beneficial to include comparisons with additional benchmarks, evaluating a wider range of models to yield deeper insights.

2. The paper lacks a detailed breakdown of the sample categories. Providing statistical information on the task categories would help demonstrate the scope and coverage of the benchmark.

3. The benchmark currently offers a relatively small set of tasks. Expanding the sample size in future iterations would improve the benchmark's applicability. Compared to benchmarks like Mind2Web and WEBLINX, the proposed benchmark’s dataset remains limited.

4. Although the authors highlight their community-driven platform and a cost-effective bi-monthly data maintenance schedule, the benchmark project appears less active. Notably, the last update was two months ago, and several long-standing issues remain unresolved.

**Questions:**

1. Will the heatmap for evaluation function accuracy across annotation steps be generated automatically upon completion of the evaluation? This could provide users with useful insights into model performance. Additionally, could some textual analysis be offered through LLM integration?

2. Are all key nodes weighted equally in the evaluation process?

3. Could you clarify why there are no cases labeled as "value include" in Figure 5?

4. A minor note: The planning prompt specifies a requirement for "JSON blob format," which seems somewhat contradictory, as JSON is typically string-based, while a blob refers to a binary object. Could you clarify this distinction?

---

> ### Author Response · Authors · 2024-11-19
> **Explanation of Dataset and Model Comparison Settings(Part 1 of 3)**
>
> Thank you for recognizing the value of our benchmark dataset, evaluation framework, and the openness of our data infrastructure. We understand that your concerns primarily revolve around the completeness of the dataset comparisons, the diversity and quantity of datasets, and our platform maintenance plan, along with some questions regarding specific details of the paper. We will address these issues one by one. If there are still any unclear points, we would be more than happy to clarify them further during the discussion.
>
> # 1. Explanation of Dataset and Model Comparison Settings
>
> We chose not to compare with other benchmarks for several reasons. The well-known web agent benchmarks previously published, such as WebArena[1] and VisualWebArena[2], are based on tasks defined in closed environments, making it difficult to replicate these tasks and their evaluation functions in live environment, thus preventing fair comparisons. While WebLinx[3] presents an important setting, it primarily focuses on multi-turn conversations, which differ from our current task setting. GAIA[4] is not specifically focused on web tasks, with relatively few tasks entirely operating within a web browser. The task definitions in AssistantBench[5] are insightful and clear, and we plan to incorporate this dataset in future iterations. However, many tasks in AssistantBench (e.g., "Which gyms near Tompkins Square Park (<200m) have fitness classes before 7am?") are more suited for parallel inference using multiple agents rather than a single agent. In these cases, each agent’s trajectory would be ideal for definition and evaluation under the current WebCanvas framework. We plan to support this kind of multi-agent system evaluation in the future, which we believe is essential for assessing truly complex web tasks. However, we do not believe this detracts from our contribution in this paper, which is the first to propose an online evaluation method for sequential web tasks. Our comparison with Mind2Web aimed to clearly demonstrate the limitations of reference-based evaluation in live environments, as explained in detail in Section 6.2.
>
> To clarify the model benchmarking results, our paper primarily focuses on benchmarking the reasoning capabilities of LLMs. Therefore, we fixed a universal agent framework (Appendix E.1) and evaluated the performance of both open-source and closed-source base LLMs under this framework (Table 4). We also analyzed the value of different components within this LLM-based agent framework (Table 12). Exploring the optimal framework is not the main focus of this paper, and the requirement for long context length for web agents limited the number of models we could test. We respectfully argue that we have tested a sufficient number of base models under this setting and have clearly demonstrated the performance differences among these models.
>
> >[1] WebArena: A Realistic Web Environment for Building Autonomous Agents, Zhou et al., 2023
> >
> >[2] VisualWebArena: Evaluating Multimodal Agents on Realistic Visual Web Tasks, Koh et al., 2024
> >
> >[3] WebLINX: Real-World Website Navigation with Multi-Turn Dialogue, Lù et al., 2024
> >
> >[4] GAIA: a benchmark for General AI Assistants, Mialon et al., 2023
> >
> >[5] AssistantBench: Can Web Agents Solve Realistic and Time-Consuming Tasks?, Yoran et al., 2024

---

> ### Author Response · Authors · 2024-11-19
> **Dataset Diversity and Quantity(Part 2 of 3)**
>
> # 2. Dataset Diversity and Quantity
>
> **Task Distribution and Domain Coverage**: To ensure that Mind2Web-Live reflects the domain distribution of Mind2Web, we followed Mind2Web's categorization and selected tasks from the following domains: **Travel**, **Shopping**, and **Entertainment**. Please note that the Info and Service domains in Mind2Web only appear in cross-domain tasks in the test set, which is why they are not included in Mind2Web-Live. We have provided statistics on the task distribution in the training and test sets across these domains, which can be found in Table 2 of our response to reviewer **mDNN**. We have included these statistics in Appendix A.3 of the manuscript. Additionally, Figure 13 and Figure 14 of the paper provide detailed statistics on Completion Rate and Task Success Rate across different websites, allowing us to evaluate Web Agent performance across various domains.
>
> **Task Quantity in the Mind2Web-Live Benchmark**: As mentioned in our response to reviewer **mDNN**, maintaining a large test set significantly increases the cost for community evaluation of web agents. For instance, testing 1,000 tasks per model with an average runtime of 1.5 minutes per task would take approximately one day and incur high API costs. Therefore, we decide to maintain a relatively small, high-quality, general-purpose foundation benchmark that retains sample diversity. Furthermore, the vision of our work is to enable community members to build evaluation sets in any vertical domain at a low cost in the future, rather than just centrally releasing a dataset. We also plan to incorporate additional high-quality benchmarks, such as AssistantBench, in future iterations, and work out better UI for community members to define their own evaluation sets more easily.

---

> > ### Author Response · Authors · 2024-11-19
> > **Explanation of Community Maintenance and Other Detailed Questions About the Paper(Part 3 of 3)**
> >
> > # 3. Explanation of Community Maintenance
> > Thank you very much for your question! We are pleased that you have noted our community contributions and code base update activities.
> > Since the release of our work, we have received 12 suggestions for code improvement from various channels (such as system feedback, email, GitHub issues, and social media direct messages). Additionally, we have received modification suggestions from 5 research groups regarding the dataset, covering a total of 25 tasks.
> > Regarding the code repository, we have updated several versions, including a more user-friendly plug-and-play showcase for evaluation module usage, updated data-related actions, released new evaluation functions focused on practicality, and added support for more models. For the dataset, we performed maintenance in August and October, adhering to our commitment of bi-monthly maintenance. To ensure continued quality, the primary authors have committed to maintaining the framework's usability and the dataset's validity over the next two years.
> > As for the open issues on GitHub, one is related to an inherent limitation of our framework, while the other two remain open because we were unsure if our responses fully resolved the reporter's needs, and we are awaiting further feedback.
> > We understand that the information on public platforms may have certain limitations, and we are happy to provide clarification on these issues. However, as a double-blind conference, ICLR states in its Reviewer Guide that it does not recommend relying on open-source information for evaluation purposes. While we are delighted by your kind interest in our community contributions, we believe that the activity level of a community is subjective (especially considering that private communications are difficult to quantify). Therefore, we kindly request that this not be used as grounds for dismissing the merits of our work.
> >
> > # 4. Other Detailed Questions About the Paper
> >
> > - **Will the heatmap for evaluation function accuracy across annotation steps be generated automatically upon completion of the evaluation? Could some textual analysis be offered through LLM integration?**
> >
> > Thank you for your valuable suggestions! Currently, the heatmap for evaluation function accuracy needs to be generated manually after the experiment completes; it is not yet fully automated. However, in our future update plans, we intend to add more analysis modules for the reasoning process and results, and to decouple these from the baseline framework, allowing users to utilize them as needed.
> > Regarding textual analysis, we greatly appreciate your suggestions. Could you share specific types of analysis you believe would be valuable for users? We will carefully consider these requirements and incorporate them into future updates as much as possible.
> >
> > - **Are all key nodes weighted equally in the evaluation process?**
> >
> > Yes, currently all key nodes are weighted equally. In Figure 12, we show the relationship between the average score of key nodes and their order, where earlier nodes tend to be easier to complete, especially the first key node, which often corresponds to navigating to the task-related website (usually through URL include match). Despite this, we have not yet found a reliable way to adaptively adjust the weights. We hope future work will address this limitation.
> >
> > - **Could you clarify why there are no cases labeled as "value include" in Figure 5?**
> >
> > Thank you for your careful review of the figure details! During data annotation, we prioritized using "URL" or "element path" to define key nodes. For functional buttons (e.g., submit buttons), if their position is fixed, we typically match using the element path; if the position is not fixed, we use "element value exact match" since the button name is generally consistent. For semantic content like product information, we use "semantic match". So far, we have not encountered a scenario where "element value include match" was required.
> >
> > - **Clarification on JSON Blob Format**
> >
> > Thank you for your detailed review, and we apologize for any confusion caused. We initially intended to refer to a "json-formatted code block" but mistakenly described it as a "JSON blob." This has been corrected in the latest version of the paper. Please note that this correction does not affect the output's accuracy, as the prompt examples and actual output remain JSON-formatted code blocks.
> >
> > # 5. Final Remarks
> >
> > We hope our response has adequately addressed your questions. If you have any further doubts or suggestions, please feel free to reach out, as your feedback is invaluable to improving the WebCanvas framework. If our explanations have resolved your concerns, we sincerely hope you might reconsider your rating of the paper. Thanks again for the review!

---

> > > ### Comment · Reviewer_BnDr · 2024-11-22
> > > **Thanks for your responses**
> > >
> > > Thank you for your detailed responses. However, my concern regarding maintenance has not been fully addressed. Could you provide a simple example that illustrates the effort required to correct a sample error caused by changes in the web environment?
> > >
> > > Regarding Question 1, the suggestion for textual analysis means that once the evaluation is complete, the benchmark will generate a heatmap of errors and analyze the results using a Large Language Model (LLM) — an optional feature. The goal is to provide insights and recommendations for further improvement of the evaluated agent. For example, the analysis will summarize which categories of tasks are more challenging for the agent and explain why the agent consistently fails in those tasks.

---

> > > > ### Author Response · Authors · 2024-11-25
> > > > **Explanation of Maintenance Effort and Error Analysis**
> > > >
> > > > # 1. Maintenance Effort
> > > > We use an automated testing system (L251) to detect whether tasks have become invalid. This system replays workflows step by step based on the annotated data to check the validity of workflows and key nodes. To minimize the risk of triggering anti-crawling mechanisms on websites, we set an interval of 20 seconds between each operation. Running the Mind2Web-Live test set in a single-threaded mode takes about **300 minutes**. Upon completion, the system generates a test report for each task (e.g., Figure 17). For tasks that fail, we manually inspect and fix them.
> > > >
> > > > During the maintenance in August, which was our first attempt, we spent about **3 hours** completing the entire process. In the review period, we conducted another maintenance session, which, thanks to community feedback, required **less than 2 hours**. We have uploaded the latest version of the dataset in the supplementary material folder for review. Below, we discuss the time and effort required for maintenance using examples of the three types of task failures illustrated in Figure 2 (left):
> > > > - **UI Change (Green Section):** This is relatively common since websites frequently update their designs over time. To improve the stability of key nodes, WebCanvas recommends using more robust URL matching as the evaluation function rather than relying on element selector paths (Element Path).
> > > >
> > > >   For example, in the task "Find the last game of the season for the Toronto Raptors on sports.yahoo.", although Yahoo Sports changed its navigation bar design from black (Mind2Web recording in 2023) to white (Mind2Web-Live recording in 2024), URL-based matching rules (e.g., `/toronto`) remained unaffected. As a result, most key nodes are still valid. However, key nodes relying on element paths, such as `#Col2-5-TeamSchedule-Proxy > div > div > div:nth-child(4) > div`, may become invalid. In such cases, we only need to manually update the selector to the correct path, without re-annotating the entire task. As mentioned in L261, this type of maintenance typically takes around **2 minutes**.
> > > > - **Workflow Change (Yellow Section):** If the workflow changes but the key nodes remain valid, maintenance is optional. However, to improve the visual presentation for community users (Figures 18, 19) and better reflect the current web environment (L255), we generally re-record the workflow and update the key node annotations. This type of maintenance, involving both re-recording and re-annotation, usually takes about **5 minutes** (L260).
> > > > - **Task Expired (Red Section):** If the website hosting the task has been shut down or is no longer available (e.g., the "Book Depository" website, which was discovered to be shut down during the Mind2Web-Live recording), we simply delete the task. Similar situations may occur in future maintenance. Removing an invalid task is straightforward and takes only **a few minutes**.
> > > >
> > > > # 2. LLM-Based Error Analysis
> > > > Thank you once again for your response and the excellent suggestions in question 1. We agree that analyzing the failure cases of agent reasoning is non-trivial, especially in a complex online environment. We, among other developers and researchers often find it challenging to manually evaluate all reasoning paths and identify model shortcomings. Both the results from a single free-form task execution and batch task execution with ground truth can benefit greatly from a model-based evaluator or analyzer that summarizes the outcomes and provides insights.
> > > >
> > > > **In the supplementary materials, we implemented a simple demo version**, and found that advanced models like GPT-4o showed promising results by providing fruitful and accurate summaries and improvement suggestions for individual task reasoning. However, we acknowledge that this is not the optimal interaction approach, and we will continue to improve the experience of this module. We believe a better experience would be one that automatically categorizes failure reasons for batch tasks, allowing developers and researchers to easily index each type of error with its corresponding task and trajectory. The updated version is under engineering and will be released soon. This would make it more efficient to improve the agent model's performance and quickly gain insights from the evaluation of tasks in an online environment.
> > > >
> > > > # 3. Summary
> > > > Thanks again for your response! While maintenance requires some time, community engagement has reduced it by about one-third. We believe that with contributions from the community and us, the Mind2Web-Live dataset will continue to evolve in a community-driven, continuously updated manner, driving the research and deployment of Web Agents in real-world applications.
> > > >
> > > > If we haven't fully addressed your concerns, we are committed to providing further clarifications during the final days of the review process. We sincerely hope that our efforts and the contributions of this work align with your expectations and merit a higher score!

---

> > > > ### Author Response · Authors · 2024-11-30
> > > > **Thank You and Looking Forward to the Discussion**
> > > >
> > > > Dear Reviewer BnDr, We wanted to say in advance our heartfelt thank you's for the time and effort you've put into both the reviews that have passed as well as the upcoming current discussion period. We know that you all have your own papers that you have to deal with during this busy time, and sincerely appreciate the time you've taken to spend on ours.
> > > >
> > > > We are so excited about this paper and its findings, so we are very much looking forward to the upcoming discussions with you! Please don't hesitate to ask us any questions big or small, and we are happy to provide any further clarifications.

---

### Official Review · Reviewer_mDNN · 2024-11-08

**Soundness:** 3
**Presentation:** 4
**Contribution:** 3
**Rating:** 6
**Confidence:** 4

**Summary:**

This paper proposes Webcanvas ,a benchmarking framework for evaluating web agents in dynamic online environments. In this framework, a new metric is proposed based on 'key nodes'. Then authors construct a online and dynamic dataset called Mind2Web-Live, which builds upon the existing Mind2set dataset. Mind2Web-Live includes extensive annotation data collected through human labor and will be regularly updated and maintained by authors. Finally, various models are evaluated on Mind2Web-Live, providing some insights according to the results.

**Strengths:**

- Introduces a innovative evaluation framework WebCanvas for web agent. By focusing on “key nodes”, this framework provides a more reliable and accurate assessment compared to traditional methods that only consider the final task success rate.
- Constructs a online and dynamic benchmark Mind2Web-Live that is an enhanced version of the original Mind2Web static dataset.
- The authors have developed a community-driven platform where users can report issues with the dataset, and regular updates are performed.

**Weaknesses:**

- When the data size was reduced from Mind2Web's original 2000+ tasks to 500 +, the authors did not analyze how many different domains the Mind2Web-Live can cover and whether there are enough tasks for each domain.
- There is a problem of scalability in this dataset because updating data requires people to maintain it. When the scale of dataset increases, maintenance costs will increase.

**Questions:**

1. Can you provide more details about the data distribution in the benchmark. Such as how many domains this benchmark can cover?  and how many task in each domain?
2. Whether the agent running in the real word environment will have a negative impact on related websites?

---

> ### Author Response · Authors · 2024-11-18
> **Authors’ Response (Part 1 of 3)**
>
> Thank you for taking the time to review our paper, and we truly appreciate your recognition of the contributions of WebCanvas!
>
> We believe that all your concerns can be addressed during the discussion phase. Please refer to our responses below. If there are any remaining unclear points, we would be more than happy to further clarify them during the discussion.
>
> # 1. On the Distribution of Data and Tasks in Mind2Web-Live
>
> **Data Annotation Process:** Given the significant time and effort required to annotate tasks and key nodes in a real-world online environment, we had to select a subset of tasks from the original Mind2Web dataset, which contains over 2,000 tasks, to construct Mind2Web-Live. To retain the characteristics of Mind2Web as much as possible, we avoided making substantial modifications to the task descriptions.
>
> **Construction of the Training Set:** From the original Mind2Web training set (comprising 1,009 tasks), we excluded all time-sensitive tasks. Subsequently, 75% of the remaining tasks were annotated with key nodes, resulting in the Mind2Web-Live training set, which includes a total of 438 tasks.
>
> **Construction of the Test Set:** The test set is based on the cross-task subset of Mind2Web test set. After removing time-sensitive and unannotatable tasks, we re-annotated the remaining tasks to form the Mind2Web-Live test set. It is important to note that the Mind2Web test set includes cross-website and cross-domain tasks, totaling over 1,000 tasks. However, annotating these tasks involves considerable costs, and large-scale online testing imposes significant time and API usage expenses. For example, testing 1,000 tasks at an average runtime of 1.5 minutes per task would take approximately 1 day and result in high API costs, which are unaffordable for many researchers. As a result, we had to forgo the annotation of this portion of the data.
>
> The selection ratio of tasks from the Mind2Web dataset for Mind2Web-Live is shown in Table 1.
>
> **Task Distribution and Domain Coverage:** We followed the same task categorization used in the Mind2Web. Tasks were selected from the following domains: **Travel**, **Shopping**, and **Entertainment**. (Note: The Info and Service domains in Mind2Web only exist in the test set's cross-domain tasks and are therefore not included in Mind2Web-Live.) We analyzed the distribution of tasks across these domains for both the training and test sets, as detailed in Table 2. Furthermore, in Figures 12 and 13 of the paper, we provide a comprehensive analysis of the Completion Rate and Task Success Rate for tasks across different websites to evaluate the Web Agent's performance in various domains and subdomains.
>
> Mind2Web-Live represents an innovative attempt to evaluate Web Agents in online environments. Although the data selection process involved certain limitations, we adhered to Mind2Web’s task categorization standards and preserved its domain distribution. The intuition behind this work is mainly the scalability of this define-then-evaluate framework for the community. Moving forward, we will collaborate with the research community to expand the live evaluation benchmarks for web agents further, bringing in brilliant data resources and thoughts like AssistantBench[1], and other vertical domains.
>
> Table 1:
> | Mind2Web Dataset | Subcategory | Retain (excluding time-based removal) |
> |-------------------|-----------------|------------------------------------------|
> | Train | | 75% |
> | Test | Cross-Task | 100% |
> | Test | Cross-Website | / |
> | Test | Cross-Domain | / |
>
> Table 2:
> | Domain | Subdomain | Mind2Web-Live Test | Mind2Web-Live Train |
> |-------------------|----------------|---------------------|---------------------|
> | Entertainment | Sports | 9 | 32 |
> || Event| 5 | 20 |
> || Game| 3 | 24 |
> || Movie| 9 | 30 |
> || Music| 5 | 18 |
> || General| 3 | 28 |
> | Shopping| Auto | 7 | 33 |
> || Department| 6 | 8 |
> || Digital| 6 | 15 |
> || Fashion| 3 | 15 |
> || Speciality | 13 | 44 |
> | Travel | General | 0 | 11 |
> || Airlines | 5 | 18 |
> || Car rental | 1 | 11 |
> || Ground | 9 | 28 |
> || Hotel | 3 | 12 |
> || Restaurant | 6 | 31 |
> || Other | 11 | 60 |
> | **Total** || **104**| **438**|
>
> > [1] AssistantBench: Can Web Agents Solve Realistic and Time-Consuming Tasks?, Yoran et al., 2024

---

> ### Author Response · Authors · 2024-11-18
> **Authors’ Response (Part 2 of 3)**
>
> # 2. On the Maintenance Costs and Community Contributions
>
> **Maintenance Costs:** It is true that as the dataset scales up, maintenance costs may increase accordingly. However, the motivation behind this project is to optimize the cost. As described in Section 4.2, we developed an automated maintenance system capable of detecting invalid workflows and key nodes to run each two months. Additionally, for issues that the system cannot detect, community members can report problems through our ```bug report``` module in our platform. This multi-channel maintenance mechanism is designed to reduce manual effort and minimizes the accumulation of errors.
>
> As for the cost of corrections, four authors collectively spent less than one hour completing a round of maintenance(L258), demonstrating that our approach is efficient and feasible at the current scale. Even as the dataset continues to expand in the future, we are confident that maintenance will remain feasible through the appropriate optimization of workflows.
>
> **The Inevitability of Key Node Evaluation:** We acknowledge that in a real-world online environment, data obsolescence is unavoidable. However, with our Key Node-based task maintenance approach, we can achieve relatively reliable and efficient management. In comparison, other common evaluation methods have notable limitations:
>
> - **Reference-based methods:** These methods rely on fixed paths, which are incapable of handling the diversity of task paths in online environments. As shown in Figure 10, the agent and the annotated task take completely different paths, yet reference-based evaluation methods fail to account for such variations, leading to reduced reliability.
>
> - **Outcome-based methods:** Evaluating tasks solely based on their final state through specific scripts execution is similarly sub-optimal for a scalable evaluation system. First, certain tasks cannot be assessed using only the final state alone. For example, consider the task of submitting a form. If we rely on a single state to determine whether the form content is valid and whether the submission is completed, there are two possible approaches: One approach is to monitor the API call for submitting the form (which may not be permitted by the website). Another approach is to check the  feedback page indicating successful submission. However, if the website does not include such a feedback design, it becomes impossible to evaluate the task. Apart from this limitation, maintaining these scripts incurs high costs, requiring specialized expertise and significant time investment, as stated in OSWorld[1], where the scripts were annotated by the authors themselves. In contrast, our Key Node-based approach, relying on URL and element validation, can be conducted by annotators with basic computer knowledge. Furthermore, in-progress key nodes evaluation is dispensable. Our findings(L373) indicate that only 47% of tasks can be effectively evaluated using final key node alone.
>
> **Regarding Community Contributions and Extensibility:** We actively encourage community participation in maintaining the dataset. Our code and community platform allow for flexibility to support community-driven contributions. Over the past few months, we have received 12 improvement suggestions for our agent code and 25 task modification requests from 5 groups through various channels. This highlights the community's recognition and engagement in our work. In the future, as the dataset scales beyond the capacity of the author team to maintain independently, we are open to hire paid annotators for data maintainance for the community to move forward.
>
> > [1] OSWorld: Benchmarking Multimodal Agents for Open-Ended Tasks in Real Computer Environments, Xie et al., 2024

---

> > ### Author Response · Authors · 2024-11-18
> > **Authors’ Response (Part 3 of 3)**
> >
> > # 3. On the Impact on Users and Websites
> >
> > **Impact on Users:** All tasks in our benchmark do not require a logged-in user state. During the annotation phase, we conducted rigorous reviews to ensure that no user privacy data would be affected, nor would any existing website data be altered. Furthermore, our evaluation methodology does not rely on user account information, eliminating potential risks of privacy breaches.
> >
> > **Impact on Websites:** The websites selected for testing are all internationally renowned platforms, such as e-commerce sites and information portals, which typically handle millions of daily active users. The tasks designed for each website involve only a small number of straightforward operations. Current Web Agents are limited in speed, executing only a few actions per minute—well below the average operational rate of human users. Consequently, the test traffic generated by Web Agents is negligible and does not significantly affect the normal functioning of these websites.
> >
> > WebCanvas evaluates web agents in live environments, providing data support and practical insights to really connect web agents to **Product Environment**. This evaluation method not only bridges the gap left by traditional static benchmarks that fail to reflect dynamic web environments but also helps developers optimize web agents to meet diverse user needs, particularly for individuals facing visual or motor impairments who rely on accessibility features. Enabling agent traffic in real websites is non-trival, however, the community needs an easy-to-use approach to access the real agent performance to define the research and product boundaries.
> >
> > # 4. Final Remarks
> >
> > We sincerely appreciate your attention and valuable feedback on our work, especially regarding concerns about data quality and the impact on the web environment! WebCanvas represents an innovative research endeavor for real-world web agents, introducing the concept of Key Node-based evaluation framework, coupled with the Mind2Web-Live dataset. We achieve a strong step forward in evaluating web agents in live environments in a in-progress scalable way—a challenge that many existing methods struggle to address. Moreover, we are actively collaborating with the community to continuously refine our codebase and update expired datasets, ensuring the long-term validity and reliability of our framework. If you have further questions or suggestions regarding our paper or response, we would be delighted to engage in further discussions and address any concerns promptly. We genuinely hope that through additional communication, we can provide greater clarity and garner your recognition of our work.
> >
> > We kindly hope that you might reconsider your score in light of these efforts and the potential impact of our work. Please let us know if you’d like any further adjustments!

---

> ### Author Response · Authors · 2024-11-30
> **Thank You and Looking Forward to the Discussion**
>
> Dear Reviewer mDNN, We wanted to say in advance our heartfelt thank you's for the time and effort you've put into both the reviews that have passed as well as the upcoming current discussion period. We know that you all have your own papers that you have to deal with during this busy time, and sincerely appreciate the time you've taken to spend on ours.
>
> We are so excited about this paper and its findings, so we are very much looking forward to the upcoming discussions with you! Please don't hesitate to ask us any questions big or small, and we are happy to provide any further clarifications.

---

### Official Review · Reviewer_uzoM · 2024-11-11

**Soundness:** 2
**Presentation:** 3
**Contribution:** 2
**Rating:** 3
**Confidence:** 4

**Summary:**

The paper is devoted to the development of new benchmark for web agents, which should demonstrate flexibility and tolerance to (1) alternative (non-canonical) trajectories of task completion and (2) dynamic nature of the web, where sites and their features constantly evolve.

The key idea of the paper is to introduce “key nodes” in the task completion process, which designate the inevitable intermediate states of requests and URLs.

**Strengths:**

The motivation and the problem are very relevant

**Weaknesses:**

The technical quality of the work is under concerns. The work relates to evaluation methodology, and the main contribution is the proposed benchmark based on key nodes. I expect an analysis of how the proposed metric for web agents correlates with the goal metrics such as success rate based on outcomes. We can annotate, for a number of agents, outcome results for a representative number of tasks, and compare the correlation between “key nodes-based success rate” and outcome-based success rate against the same correlation for “step-based success rate” proposed in Deng et al., 2024.

This is a common requirement for new metrics in methodology papers: to look at the directionality, see e.g. “Using the Delay in a Treatment Effect to Improve Sensitivity and Preserve Directionality of Engagement Metrics in A/B Experiments” by Drutsa et al.

Table 3: the result itself is expectable, because mindAct is based on direct finetuning to ground-truth actions, which are then used for evaluation of success rate in the offline setting. Such approach makes MindAct less generalizable to flexible metric and dynamic environment, unlike GPT-3,5/4, which are used with in-context learning, without finetuning.

**Questions:**

Why GPT-3,5/4 perform better in the more challenging online setting as compared to offline setting (Table 3)? It is not clear what was the protocol for online setting in this table. Authors only said that “evaluation metrics … differ” and, in online setting, “we evaluate the intermediate state, not the referenced action”. Is this metric exactly “Task Success Rate” described in Section 3.2?

---

> ### Author Response · Authors · 2024-11-19
> **Comparison of Key Node Evaluation with Other Evaluation Methods in Online Environments**
>
> Thank you for taking the time to review our paper and for acknowledging the motivation behind WebCanvas!
>
> We believe that all your concerns can be addressed during the discussion phase. Please refer to our responses below. If there are any remaining unclear points, we would be more than happy to further clarify them during the discussion.
>
> # 1. Comparison of Key Node Evaluation with Other Evaluation Methods in Online Environments
>
> **Comparison with Reference-Based (or Step-Based) Evaluation**
> In Section L368-L371 and Figure 10, we demonstrated that reference-based evaluation does not work effectively in complex online environments with agent tasks. By visualizing both human-annotated and agent-executed trajectories, we found that even when all key nodes are satisfied, there can be significant differences in possible trajectories. However, reference-based evaluation methods fail to account for such variations, leading to reduced reliability. The experiments comparing Key Node-based evaluation with reference-based evaluation also show that results from step-based testing on offline datasets do not necessarily reflect model performance in live online environments (please see Table 3 and the description of the second part comment below).
>
> **Comparison with Outcome-Based Evaluation**
> Thanks for your question to help us clarify this comparison further. Outcome-based evaluation is insufficient for online tasks where completion cannot be judged solely by the final outcome state. For instance, in a form submission task, relying on a single state to determine whether the form content is valid and whether the submission is completed has challenges. One approach is to monitor the API call for form submission, which may not be allowed by the website, while another is to check if a feedback page appears. However, if there is no feedback design, evaluation becomes impossible. Moreover, creating such scripts is time-consuming, requiring specialized expertise and significant time investment, as highlighted in OSWorld[1], where scripts were manually annotated by the authors. Additionally, the dynamic nature of online environments necessitates constant script maintenance, further increasing costs, thus it's suboptimal in a scalable evaluation system and complex domain like web environment. In contrast, our Key Node-based approach, which uses URL and element validation, can be conducted by annotators with basic computer knowledge. We will include this discussion in Section 6.2 of the paper. Evaluating tasks solely by the final key node is also suboptimal, as shown in our findings (L373), which indicate that only 47% of tasks can be effectively evaluated using the final key node alone. The Key Node-based evaluation also provides additional benefits, such as evaluating agent progress throughout the task, which facilitates analyzing and improving agent performance. This approach also enabled us to conduct experiments on planning with improved in-progress rewards, demonstrating the value of better reward signals for in-context reasoning.
>
> We hope this illustration highlights the completeness and accuracy of key node-based evaluation, making it a superior method for assessing online agent performance.
>
> > [1] OSWorld: Benchmarking Multimodal Agents for Open-Ended Tasks in Real Computer Environments, Xie et al., 2024

---

> > ### Author Response · Authors · 2024-11-19
> > **Clarification on Insights from Online & Offline Experiments**
> >
> > # 2. Clarification on Insights from Online & Offline Experiments
> > Thanks for bringing up this question to help us emphasize the experiments further. As mentioned in L300 of the paper, we ensured that the input prompts for MindAct were consistent in both online and offline environments. We also maintained consistent input prompts for MindAct and GPT-3.5/GPT-4. In the offline environment, MindAct was evaluated using a step-based method due to the inherent limitations of the evaluation data, which made it impossible to fully align the evaluation settings between online and offline scenarios. However, we can still observe the relative performance relationship: models trained on the Mind2Web training set tend to overfit to the evaluation set and method. We provided multiple sets of data to demonstrate the robustness of this relative relationship. In L314 of the paper, we conducted an initial analysis where we annotated 100 evaluation steps of GPT-4 and found that approximately 30% of its actions were reasonable, further demonstrating the limitations of step-based evaluation. Additionally, in L316-L319, we performed a qualitative analysis of MindAct's performance in the online environment and found that the model frequently encountered difficulties in recovering from erroneous states, further indicating overfitting. Our goal is to first quantitatively demonstrate that step-based testing on offline datasets does not necessarily reflect the model's real-world performance in online environments, which also partially addresses the first question about evaluation method comparison.
> >
> > As for the evaluation metrics used in the online and offline experiments, we applied the same task success rate metrics defined in this work, where a task is considered successful if all designated key nodes are achieved. Thank you for pointing this out—we will provide further clarification in L306.
> >
> > # 3. Summary
> > WebCanvas focuses on creating a more scalable and optimized evaluation system to truly connect web agents to live environments and assess their capabilities. This includes introducing new metrics and providing a set of open-access data infrastructures to the community. We have also provided a foundational benchmark based on this framework, along with thorough experiments to evaluate the performance of current web agents in live settings.
> > Although there are still some limitations, as outlined in L470-L475, we have made a steady step forward towards scalable live evaluation for web agents. We are also committed to collaborating with the web agent community to maintain and further improve our framework and data. We sincerely value your feedback and encourage any further discussion—whether it involves additional explanations, experiments, or other suggestions. We genuinely hope you will reconsider the score of our work in light of these efforts and its potential impact. Please let us know if you need further clarificaitons!

---

> ### Author Response · Authors · 2024-11-30
> **Thank You and Looking Forward to the Discussion**
>
> Dear Reviewer uzoM, We wanted to say in advance our heartfelt thank you's for the time and effort you've put into both the reviews that have passed as well as the upcoming current discussion period. We know that you all have your own papers that you have to deal with during this busy time, and sincerely appreciate the time you've taken to spend on ours.
>
> We are so excited about this paper and its findings, so we are very much looking forward to the upcoming discussions with you! Please don't hesitate to ask us any questions big or small, and we are happy to provide any further clarifications.

---

### Author Response · Authors · 2024-12-04
**Summary of Rebuttal**

# Summary of Rebuttal:

In the individual Reviewer questions below, the Reviewers brought up a variety of interesting issues that have helped us to think more deeply about this approach. Motivated by these points, we provided detailed rebuttal and some supplementary materials. Happily, we found that all the worries that the Reviewers brought up were thankfully, mitigatable and clarifiable with relative ease. Our new illustrations and supplementary materials are summarized and listes as below:

- Further elaboration on our methods, including detailed comparisons between key node based evaluation, outcome-based evaluation, and reference-based evaluation. We have also clarified the results of the online vs. offline experiments.

- A more detailed explanation of our dataset distribution, including statistical breakdowns and our data selection methodology.

- An in-depth explanation of the maintenance costs and future plans, including time investment, specific maintenance actions, and real examples.

- We have also added a demo module for summarizing agent performance, aimed at improving the accessibility of batch evaluation results and enhancing agent optimization efficiency.

- Detailed explanations and revisions have been made for specific points of concern regarding the article’s phrasing.

We would also like to emphasize the central, **novel contributions** of our work:

- Our work is the first to systematically address the testing challenges of web agents in real-world online environments, a key step toward practical applications.

- We introduced an advanced evaluation method, **key node evaluation**, and made a set of tools available to help web agent community define web agent test sets and conduct robust testing. Additionally, we expanded the well-known **Mind2Web** dataset with an online version, **Mind2Web-Live**, based on the key node evaluation approach.

- Using the **Mind2Web-Live** dataset, we benchmarked the performance of various foundational models on online web tasks on different live websites and network conditions and, for the first time, discussed the differences in evaluation methods for online web task evaluation.

- Our experiments demonstrated the importance of process feedback quality for **in-context reasoning**.

All in all, we believe that this work provides noval and important data pipelines and resources for web agent/UI agent community to define and test agent performance with ease. This work also provides novel robust predictive results, and conprehensive analysis towards different evaluation methods and experiment settings for live web agent evaluation, and would benefit the ICLR community in this venue. We hope that the reviewers agree, particularly given our new added illustrations and supplementary materials and see fit to raise their scores.

---

### Note · Authors · 2024-12-15

I have read and agree with the venue's withdrawal policy on behalf of myself and my co-authors.